# Intermittent fasting attenuates glial hyperactivation and photoreceptor degeneration in a NaIO$_3$-induced mouse model of age-related macular degeneration

Jingzhen Li [1,2,3], Beibei Wang[1,2,3], Pinjie Liu[1,2], Xuecheng Qiu[1,2], Qiyun Bian[1,2], Congxin Shen[1,2], Yanyan Li[1,2], Mengwen Shao[1,2] & Meng Li [1,2] ✉

Age-related macular degeneration (AMD) is the leading cause of irreversible vision loss, with limited treatments available. Recent studies suggest intermittent fasting (IF) may offer neuroprotective benefits for aging and age-related disorders, but its efficacy in AMD has not yet been established. Here, using a sodium iodate (NaIO$_3$)-induced AMD model in male mice, we find that pretreatment with an IF diet regimen mitigates NaIO$_3$-induced cellular damage and loss of both retinal pigment epithelium (RPE) and photoreceptors. Visual function tests indicate that IF preserves vision in NaIO$_3$-treated mice. Transcriptome analyses show IF counteracts NaIO$_3$-induced transcriptional dysregulation, affecting genes related to reactive oxygen species (ROS), inflammation, and photoreceptor structure. Further experimental results confirm that IF effectively reduces ROS levels and inhibits the activation of microglia and Muller cells in the retina. Collectively, these findings indicate that IF reduces ROS production and inflammation in NaIO$_3$-induced retinal damage, providing a potential therapeutic strategy for oxidative stress-induced retinal degenerative diseases, including AMD.

Age-related macular degeneration (AMD) is an increasingly prevalent visual disorder, which shows an increasing incidence with age[1,2]. AMD primarily affects the macular region, leading to progressive loss of the retinal pigment epithelium (RPE) and photoreceptors[3,4]. AMD manifests as one of two variants,dry (atrophic or non-exudative) and wet (exudative). Dry AMD accounts for approximately 90% of all AMD cases, among these, about 10-20% of patients progress to geographic atrophy (GA), the advanced stage characterized by slowly progressive atrophy of the RPE accompanied by degeneration of the adjacent neural retina[5,6]. Although the pathogenesis of dry AMD is not fully understood, it is widely accepted that chronic oxidative stress, inflammation from aging, and other adverse factors gradually destroy the RPE layer, leading to the death of nearby photoreceptor cells[7,8].

Visual processing by the retina is highly oxygen-demanding, and high metabolic activity can result in the production of significant amounts of reactive oxygen species (ROS)[9,10]. While ROS plays a role in normal cellular signaling, excessive ROS levels can lead to oxidative damage to cells. Oxidative damage by ROS, including the destruction of mitochondria, proteins, and DNA, forms the pathological basis of various retinal diseases, including AMD[11–14]. Inflammation also plays an important role in the pathogenesis of various retinal degenerative conditions[15]. Overactivation of microglia, the primary immune cells in the retina, is considered a key contributor to AMD[16]. Microglial activation precedes photoreceptor cell death in AMD[16], which may be attributed to the release of inflammatory cytokines by activated microglia. Interleukin-1β, a key mediator of innate immune responses, has been implicated in several retinal disorders, including AMD[17]. These findings highlight the urgent need to explore anti-inflammatory and antioxidant strategies for AMD management.

Mounting evidence supports the role of a healthy lifestyle, including exercise and dietary intervention, in enhancing immunity and antioxidant

[1]Jiangsu Key Laboratory of Brain Disease Bioinformation, Xuzhou Medical University, Xuzhou, Jiangsu, 221000, China. [2]Department of Biochemistry, School of Basic Medical Sciences, Xuzhou Medical University, Xuzhou, Jiangsu, 221000, China. [3]These authors contributed equally: Jingzhen Li, Beibei Wang. ✉e-mail: limeng@xzhmu.edu.cn

capacity[18–21]. Among the various types of dietary restrictions, intermittent fasting (IF) has received growing attention owing to its potential to protect against various systemic diseases, including metabolic, cardiovascular, and neurodegenerative diseases, particularly Alzheimer's and Parkinson's diseases[22,23]. IF exerts robust neuroprotective and neuroregenerative effects against nervous system impairments by suppressing the activation of the deleterious pathways associated with oxidative stress and inflammation[24–27]. Although the benefits of IF on brain health have received extensive attention, its impact on retinal pathology, particularly in AMD, has been relatively neglected. Recently, a population-based association study found that intermittent fasting by skipping breakfast correlates with a significantly reduced risk of AMD[28]. However, it is unclear whether IF prevents visual deficit and retinal degeneration and whether ROS and glial activation underlie the effects of IF on AMD.

Hence, the present study explored the effects and underlying mechanisms of a prior IF dietary management on a widely used $NaIO_3$-induced mouse model of AMD. A single administration of $NaIO_3$ induces oxidative damage and cell death within the RPE followed by the degeneration of photoreceptor cells, mimicking the symptomatic progression of AMD patients[14]. We present, for the first time, evidence that IF preserves visual function in $NaIO_3$-treated mice, as assessed using an optomotor response test. IF effectively alleviates the degeneration and loss of RPE and photoreceptor cells induced by $NaIO_3$. Furthermore, our results suggest that IF inhibits glial cell activation and excessive production of ROS in the retina of $NaIO_3$-treated mice. Additionally, experiments in middle-aged and elderly mouse models show that IF initiated at later life stages still confers resistance against $NaIO_3$-induced retinal damage. Collectively, these findings identify IF as a potential therapeutic strategy for ameliorating oxidative stress-mediated retinal degeneration.

## Results

### IF ameliorates damage in visual function in a $NaIO_3$-induced AMD model

2-month-old C57BL/6 J mice were randomly assigned to either the ad libitum (AL) or intermittent fasting (IF) groups (Fig. 1A). People practicing IF diet often observe weight loss and undergo metabolic shifts from utilization of glucose to ketones, manifested as lower blood glucose and elevated plasma ketones[29]. In consistent with the physical and metabolic changes in humans, following a 12-weeks dietary regimen, the IF group of mice exhibited reduced total food consumption and weight gain (Fig. 1B–D), as well as metabolic adaptations including lower blood glucose and elevated levels of total ketones and the major type of blood ketone β-hydroxybutyrate (β-HBA) (Fig. 1E–H). These results indicate the successful establishment of a mouse model of IF intervention.

To determine whether intermittent fasting (IF) confers protective effects against retinal degeneration, we employed a $NaIO_3$-induced model in male mice. This approach minimized hormonal confounders and aligned with standardized AMD modeling protocols[30]. $NaIO_3$ was administered via intraperitoneal injection following completion of the dietary regimen (Fig. 1A). By examining retinal damage in mice exposed to various concentrations of $NaIO_3$, we determined that an intraperitoneal injection of 25 mg/kg $NaIO_3$ induced pronounced damage to the central region of the retinal outer nuclear layer (ONL), with moderate damage to the middle and peripheral regions (Fig. S1). The spatial pattern of $NaIO_3$-induced retinopathy aligns with established models of oxidative retinal damage[30]. To determine the effects of IF on $NaIO_3$-induced visual impairment in mice, we further conducted functional studies evaluating optomotor responses (OMR), a robust technique for the examination of visual acuity (Fig. 1I, Fig. S2A)[31,32]. Under dark conditions, the $NaIO_3$-exposed mice exhibited a significant reduction in head-tracking frequency and visual acuity. IF significantly alleviated the visual deficit (Fig. 1J, K). Similar results were obtained under light conditions (Fig. S2B–C). These results indicate that IF exerts a protective effect on visual function in the $NaIO_3$-induced mouse model of AMD.

### IF attenuates $NaIO_3$-induced structural and functional disorders in the RPE

Considering that RPE degeneration precedes photoreceptor loss in the pathogenesis of AMD[1], we wondered whether IF may reduce the damage to the RPE layer in the AMD model. RPE flat-mount preparations showed marked pigment aggregation in the AL+$NaIO_3$ group compared to PBS controls (Fig. S3A–C). Histological cross-sections further demonstrated abnormal pigment fragmentation along the RPE-photoreceptor interface (Fig. S3D). Notably, IF intervention attenuated these pathological manifestations (Fig. S3A–D). Subsequently, cell morphology and density of PRE were assessed using phalloidin and DAPI staining, respectively, to visualize F-actin and nuclei in RPE whole mounts. Consistent with previous reports, $NaIO_3$ exposure resulted in pronounced morphological distortion and the loss of RPE cells, particularly in the central regions (adjacent to the ONH)[33] (Fig. 2A, Fig. S4A–C). The hexagonal structure of RPE cells was almost lost in the central region, and was markedly distorted in the middle and peripheral regions of $NaIO_3$-induced mice (AL + $NaIO_3$ group). IF drastically reduced the morphological distortion and cell loss of the RPE, especially in the central regions (Fig. 2A-C, Fig. S4A–C). Consistent with this morphological distortion, Western blotting analysis revealed that IF effectively mitigated the $NaIO_3$-induced reduction in the expression of RPE65, a specific marker for RPE (Fig. 2B, D). Phagocytosis and clearance of photoreceptor outer segments (POS) by the RPE underlie its ability to exert neuroprotection[34,35]. To evaluate the phagocytic function of the RPE, peanut agglutinin (PNA) was employed to mark the POS engulfed by RPE cells, which were labeled by TJP1 (Fig. 2E). $NaIO_3$ treatment induced a significant downregulation of PNA expression in retinal pigmfigent epithelial (RPE) cells (p < 0.01 vs. control), correlating with impaired phagocytic capacity for POS clearance. Conversely, IF intervention slightly restored POS internalization efficiency in RPE cells, as evidenced by quantitative analysis of phagocytosed POS signals (Fig. 2E-G). Lipid digestion is another key function of RPE cells, which is responsible for digesting engulfed POS membranes to maintain visual function and retinal health. Deficits in lipid digestion can lead to the accumulation of lipid droplets in RPE cells[35]. Nile red staining revealed that $NaIO_3$ treatment led to a significant increase in lipid droplet accumulation within the RPE, indicating that the processing of engulfed POS by RPE was disrupted in the $NaIO_3$-induced retinal degeneration model. The disruption of lipid digestion was alleviated by the IF regimen (Fig. 2F-H). These results indicate that IF protects RPE cells from $NaIO_3$-induced morphological and functional abnormalities.

### IF rescues $NaIO_3$-induced photoreceptor degeneration

Degeneration of photoreceptors due to dysfunctional and dead RPE cells threatens the visual functions of patients with advanced AMD[3]. Therefore, we evaluated the preservation of photoreceptor cells by the IF regimen. In $NaIO_3$-treated mice, H&E staining of the retinal sections revealed a pronounced disintegration of photoreceptor layers. Quantitative analysis showed a marked reduction in the thickness of the outer nuclear layer (ONL). The reduction in ONL thickness was more severe in the central region adjacent to the optic nerve head than that in the middle or peripheral regions (Fig. 3A, B). The inner nuclear layer (INL) also showed significant but mild thinning in the AL group (Fig. 3A, C). These results are consistent with the pathological features of patients with AMD, that exhibit degeneration of the RPE and photoreceptors, with the central regions and ONL being more severely affected[3]. IF treatment greatly rescued the $NaIO_3$-induced thinning of the ONL and INL, thereby conferring protective effects on the photoreceptors (Fig. 3A–C, Fig. S5A, B). Immunostaining of the outer segment structures of cone and rod photoreceptors with PNA and RHO, respectively, showed a dramatic decline in the density of the cone and rod outer segments in $NaIO_3$-treated AL mice, indicating that the remaining photoreceptors were degenerative. This decline was prevented by IF intervention, thus demonstrating a protective role of IF in photoreceptor degeneration (Fig. 3D–G).

To further determine the efficacy of IF treatment in mitigating $NaIO_3$-induced photoreceptor degeneration, we performed the TUNEL assay to

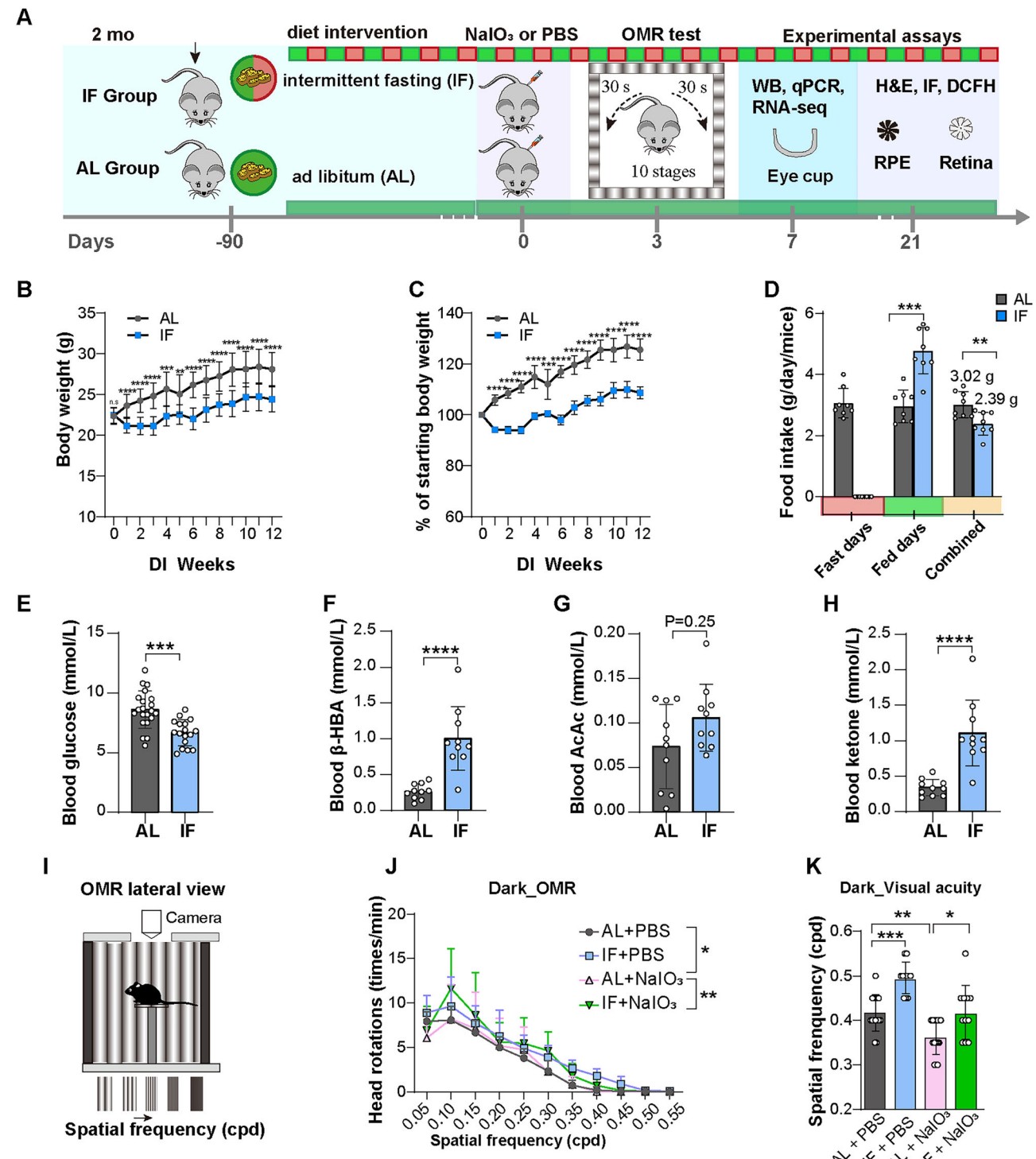

**Fig. 1 | Effects of IF on physiological parameters and visual function in a NaIO₃-induced mouse model of AMD.** **A** Schematic diagram of experimental design and assays performed in 2-month-old mice. **B** Results of weekly body weight measurements throughout the dietary intervention; $n = 30$ in AL group and 15 in IF group. **C** Percentage change in body weight during fasting compared to the beginning. **D** Average daily food consumption per mouse during intervention period. $n = 8$. **E** Fasting blood glucose levels in mice at the end of a 12-week dietary intervention. AL group, $n = 21$; IF group, $n = 17$. The concentrations of β-hydroxybutyrate (β-HBA) **F**, Acetoacetic acid (AcAc) **G**, and total ketone (**H**) in the serum of mice. $n = 10$ mice/group. **I** Schematic of the lateral view of the OMR test. **J** Quantification of the optomotor responses as the number of head rotation/min measured after dark adaptation. $n = 15$ in AL + PBS, AL+NaIO₃ group and 11 in IF + PBS, IF+NaIO₃ group. cpd, cycle per degree. Two-way ANOVA was used to analyze differences in the average number of head movements across stages. **K** The grating density corresponding to the highest visual sensitivity (visual acuity) in mice after dark adaptation. Data are expressed as mean ± SD. $P$ values were calculated using Two-way ANOVA. **$P < 0.01$, *$P < 0.05$.

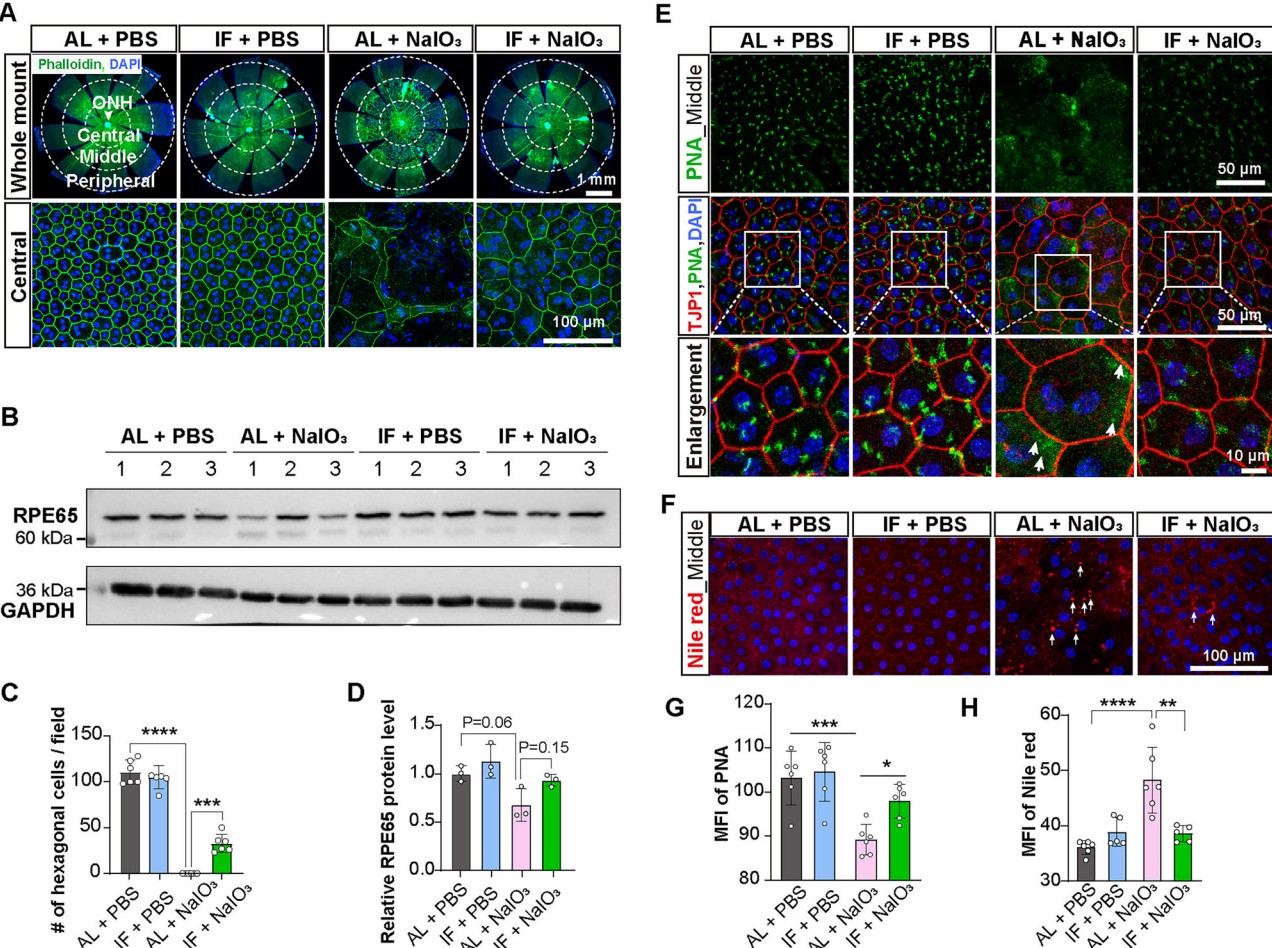

**Fig. 2 | IF protects the RPE from structural and functional disorders induced by NaIO₃.** **A** RPE morphology was assessed using phalloidin staining for F-actin and DAPI staining for nucleus on whole mounts of the RPE from 2-month-old mice. Scale bars, 1 mm (top row) or 100 μm (bottom row). **B** Western blotting analysis of RPE65, a specific RPE marker. GAPDH was used as a loading control. **C** Quantification of normal hexagonal cells within different RPE regions. $n = 6$ mice/group. **D** Quantification of protein expression levels by grayscale scanning. $n = 3$ mice/group. **E** Immunostaining of RPE by TJP1 and POS by PNA highlighting the

phagocytosis of POS by the RPE in the middle region of the RPE flat mounts. Scale bars, 50 μm (top two rows) or 10 μm (bottom row). **F** Nile red staining indicating the accumulation of lipid droplets in RPE cells. Scale bar, 100 μm. **G** Mean fluorescence intensity (MFI) of PNA staining. **H** MFI of Nile red staining. $n = 5–6$ mice/group. Abbreviations: ONH, optic nerve head. Data are expressed as the mean ± SD, $P$ values were calculated using two-way ANOVA with Tukey HSD post hoc tests in **C, D, G, H**. ****$P < 0.0001$, ***$P < 0.001$, **$P < 0.01$.

---

examine the death of retinal photoreceptor cells. Post-NaIO₃ administration, TUNEL-positive signals were observed predominantly in the ONL of AL mice (Fig. 3H, I), which is consistent with our previous observation that the ONL is the most significantly degenerated nuclear layer (Fig. 3A, B). IF treatment substantially decreased TUNEL positivity, indicating that IF ameliorated NaIO₃-induced death of retinal photoreceptor cells (Fig. 3H, I). The effects of IF on retinal cell death were further verified by the expression profiles of the BAX protein, immunostaining for cleaved caspase-3, and the transcriptional patterns of apoptosis- and necrosis-related genes (Fig. 3J, K, Fig. S6A–D). Overall, our results indicate that IF attenuates NaIO₃-induced photoreceptor degeneration and retinal damage.

### IF attenuates NaIO₃-induced oxidative stress in the retina

Next, we performed a transcriptomic analysis by RNA-seq to explore the mechanisms underlying the effects of IF on alleviating NaIO₃-induced RPE abnormalities and retinal degeneration (Fig. 4A). As shown in the PCA plot, in PBS-treated conditions, transcription profiles of IF mice were similar to those of AL mice, indicating a mild effect of IF diet on healthy mice. NaIO₃-treated mice on AL diet presented marked deviation of gene expressions from PBS-treated mice. In contrast, IF diet was able to shift gene expressions back towards PBS control mice (Fig. S7A). NaIO₃ is known to induce

cellular oxidative stress. Therefore, we first focused on the changes in oxidative stress-related genes. Gene set enrichment analysis (GSEA) revealed that NaIO₃ robustly induced genes involved in reactive oxygen metabolic processes (normalized enrichment score [NES] = 1.47) (Fig. 4B), which were suppressed by the IF intervention (IF + NaIO₃ vs. AL + NaIO₃, NES = -1.49) (Fig. 4C). We further measured the transcription levels of antioxidant enzymes previously reported to be associated with oxidative stress responses[36]. Of the ten oxidative stress-related genes tested, nine were significantly induced in the eye cups of AL + NaIO₃ mice, six of which were significantly suppressed by IF intervention, while others also showed a downward trend in their expressions (Fig. 4D).

We then assessed the changes in levels of reactive oxygen species (ROS) in the retina using the molecular probe H2DCFDA. H2DCFDA staining of retinal sections indicated that NaIO₃ exposure significantly increased ROS production, which was diminished by IF (Fig. 4E, F). Additionally, the protein levels of NADPH oxidase 2 (NOX2), which is involved in superoxide generation, were found to be markedly increased in the retinal sections following NaIO₃ treatment. This increase was attenuated in the IF group (Fig. S7B–E). Elevated ROS causes oxidative damages to DNA and increases γH2AX in cells[37]. Therefore, we performed γH2AX immunofluorescence to assess oxidative damage in the

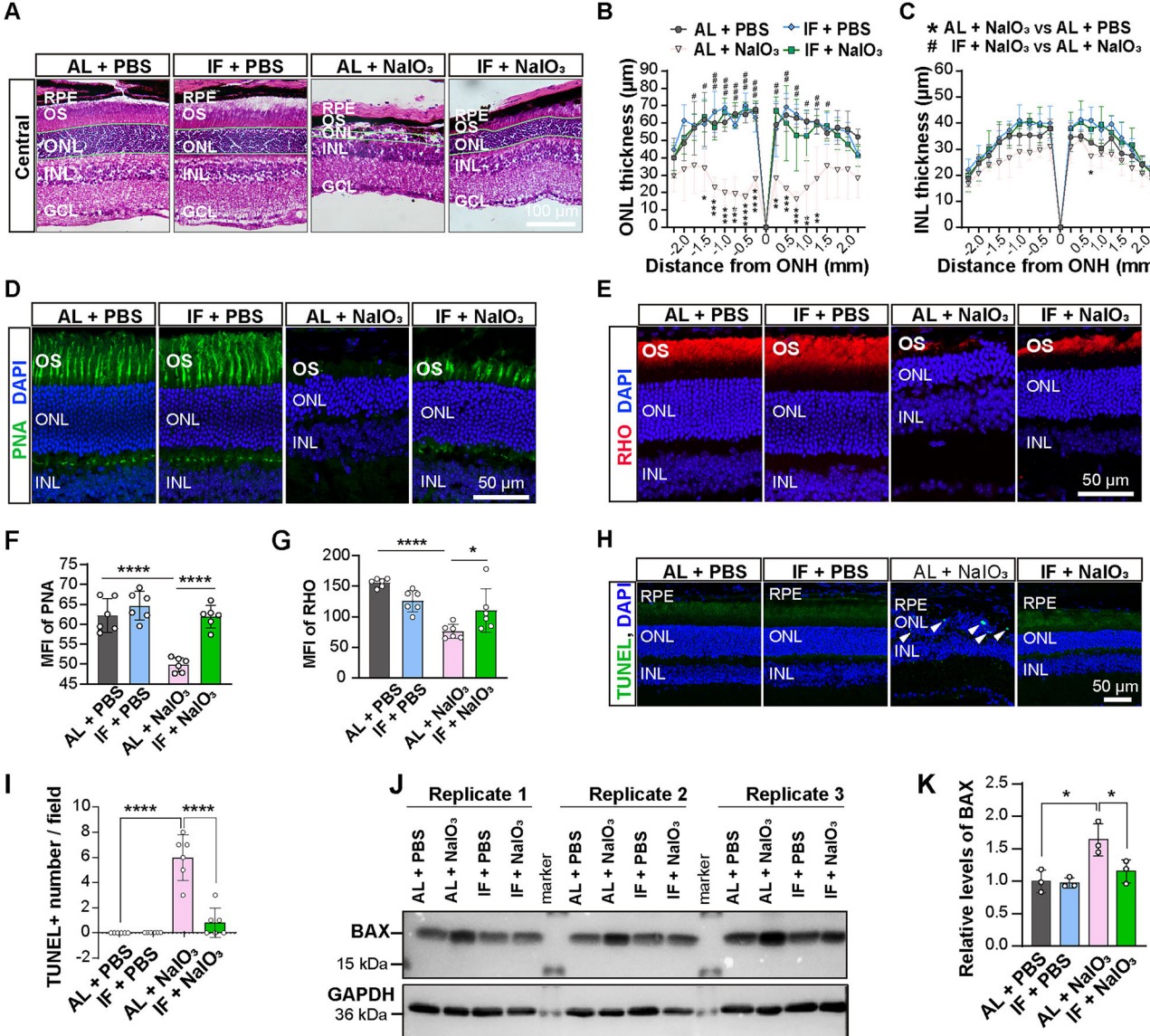

**Fig. 3 | IF rescues NaIO₃-induced photoreceptor degeneration. A** Representative images of hematoxylin and eosin (H&E) staining of the retinal sections. RPE, retinal pigment epithelium; OS, outer segment; ONL, outer nuclear layer; INL, inner nuclear layer; GCL, ganglion cell layer. Scale bar, 100 μm. **B, C** Quantification of the ONL and INL thickness at varying distances from the optic nerve head (ONH). $n = 4$ in AL + PBS and $n = 5$ in other groups. **D** Representative images of PNA labeling the POS of the cones in retinal sections. Scale bar, 50 μm. **E** Representative images of RHO labeling the POS of rods in retinal sections. Scale bar, 50 μm. **F, G** Quantification of mean fluorescence intensity of PNA and RHO. $n = 6$ mice/group. **H** TUNEL assay identifying apoptotic cells in retinal sections across groups. Scale bar, 50 μm. **I** Quantification of the TUNEL fluorescence intensity. $n = 6$ mice/group. **J** Western blotting analysis of the BAX levels, GAPDH as a loading control. **K** Quantification of the BAX levels by grayscale scanning. $n = 3$ mice/group. Data are expressed as the mean ± SD, *P* values were calculated using two-way ANOVA with Tukey HSD post hoc tests in **F, G, I, K**. ****$P < 0.0001$, ***$P < 0.001$, *$P < 0.05$.

retina. Consistent with the ROS results, NaIO₃-treated retinas exhibited a marked increase in γH2AX-positive signals in the ONL, suggesting oxidative damage in the retinal cells (Fig. 4G, H). IF intervention significantly reduced γH2AX labeling in the ONL, indicating its protective effect against oxidative damages in the retina (Fig. 4G, H). These results indicated that IF treatment reduced NaIO₃-induced ROS production and ameliorated oxidative damage to the retina.

**Transcriptome analysis revealed the inhibitory effects of IF on retinal degeneration and inflammatory pathways**
To explore additional protective mechanisms of IF, we performed further functional enrichment analysis of the RNA-seq data. The analysis revealed 1970 differentially expressed genes (DEGs) when comparing the IF + NaIO₃ and AL + NaIO₃ groups. Of these DEGs, 537 (27.3%) were up-regulated and 1433 (72.7%) were down-regulated (Fig. 5A). Analysis

of top DEGs and Gene Ontology (GO) revealed that up-regulated genes were enriched in photoreceptor cilia (e.g, *Krt23, Aspm, Kifc5b*) and visual perception (e.g, *Abca13, Serinc4*), which was consistent with our findings that IF rescues photoreceptor damage and visual deficit in the NaIO₃–induced modle (Fig. 5B, C) Down-regulated genes were significantly enriched in pro-inflammatory pathways, including those involved in cytokine production (e.g., *Spn, Il21, Bcl3, Serpinb7*), external stimulus response (e.g., *Ccl2, Ccl3, Ccl7, Klrb1b*), and immune system regulation (e.g., *Cd244a,Ccdc194*) (Fig. 5B, C). Next, we performed qPCR to validate the expressions of common cytokines, which showed that NaIO₃ activated both pro- and anti-inflammatory transcripts in the AL group, whereas IF treatment inhibited the activation of these genes (Fig. 5D, E). These findings suggest that IF treatment prevents retinal degeneration by modulating inflammation to protect the retina from oxidative stress damage.

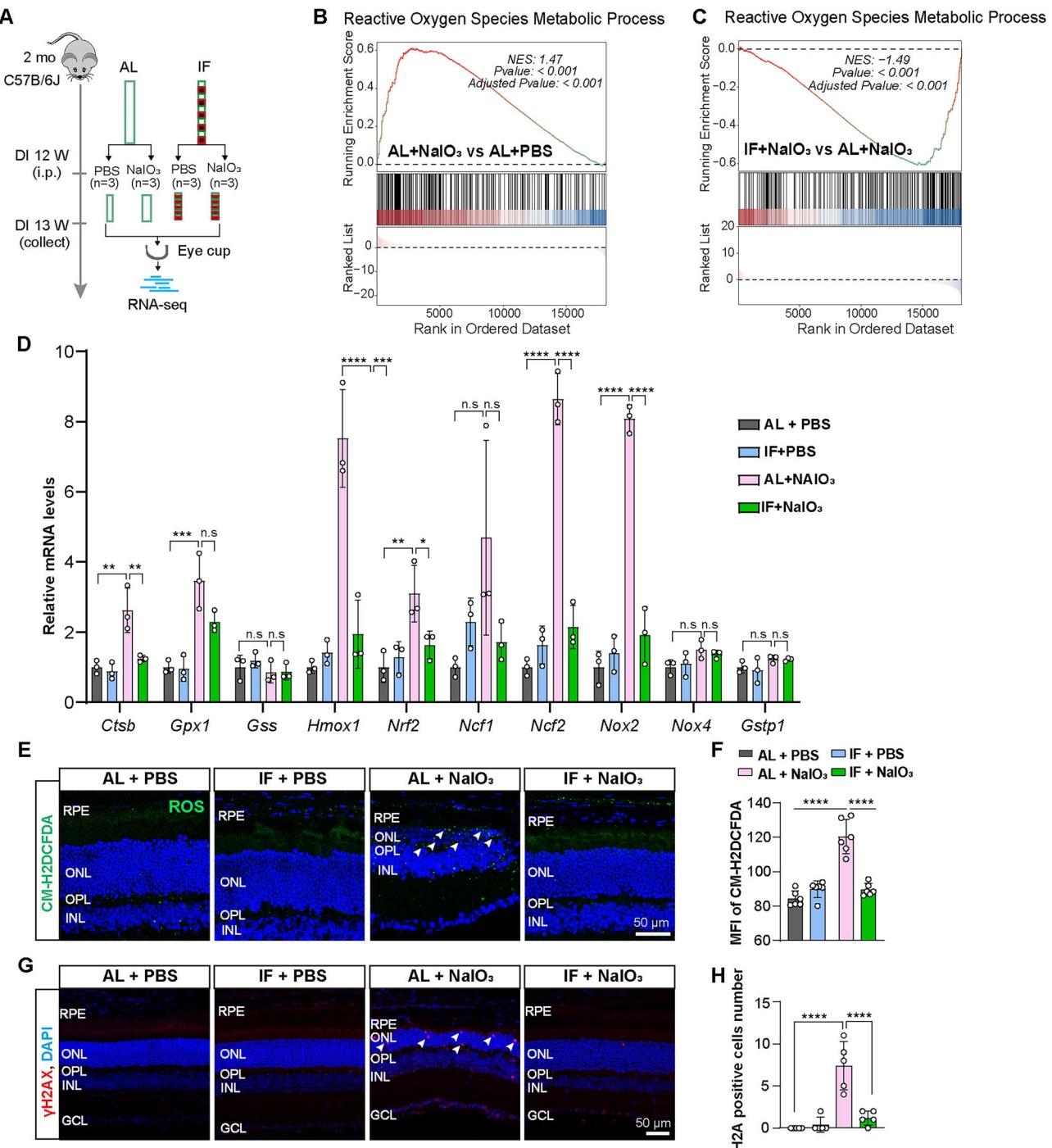

**Fig. 4 | IF attenuates NaIO₃-induced oxidative stress in the retina. A** Diagram illustrating mouse treatment and RNA-seq. **B** The GSEA profile demonstrating the significant enrichment of gene sets associated with the ROS pathway in AL+NaIO₃ mouse retinas compared to AL + PBS retinas. **C** The GSEA profile demonstrating significant enrichment of gene sets associated with the ROS pathway in IF+NaIO₃ mouse retinas compared with AL+NaIO₃ retinas. **D** qPCR analysis of the mRNA levels of oxidative stress-related genes. *Gapdh*, *Actb*, *Ubc* were used as reference genes, $n = 3$ mice/group. **E** ROS levels in the retinas of each group, as revealed by the CM-H2DCFDA probe assay. Arrows indicate ROS signals. Scale bar, 50 μm. **F** Quantification of ROS positive mean fluorescence intensity. $n = 6$ mice/group. **G** Results of γH2AX immunostaining performed on the mouse retinas after NaIO₃ or PBS administration. Scale bar, 50 μm. **H** Quantification of γH2AX positive cells number. $n = 5$ mice/group. Data are expressed as the mean ± SD, *P* values were calculated using Two-way ANOVA with Tukey HSD post hoc tests in **D, F, H**. ****$P < 0.0001$, ***$P < 0.001$, **$P < 0.01$, *$P < 0.05$.

## IF suppressed glial hyperactivation induced by NaIO₃

Our RNA-seq analysis revealed that IF inhibited NaIO₃-induced inflammatory pathways and the production of inflammatory cytokines, suggesting that IF intervention may reduces glial cell activation or infiltration in the retina, which have been implicated in the pathogenesis of AMD[15,16]. Therefore, we evaluated whether IF suppressed the hyperactivation of Müller cells and microglia. Immunostaining of retinal sections for the microglia cell marker IBA1 revealed a significant increase in IBA1⁺ cells in the retina of AL+NaIO₃ mice (Fig. 6A, B), suggesting proliferation of microglia. The microglia were also positive for CD68, a lysosomal marker of activated microglia[38], indicative of hyperactivation of microglia in AL+NaIO₃ mice. NaIO₃-stimulated increase and activation of microglia

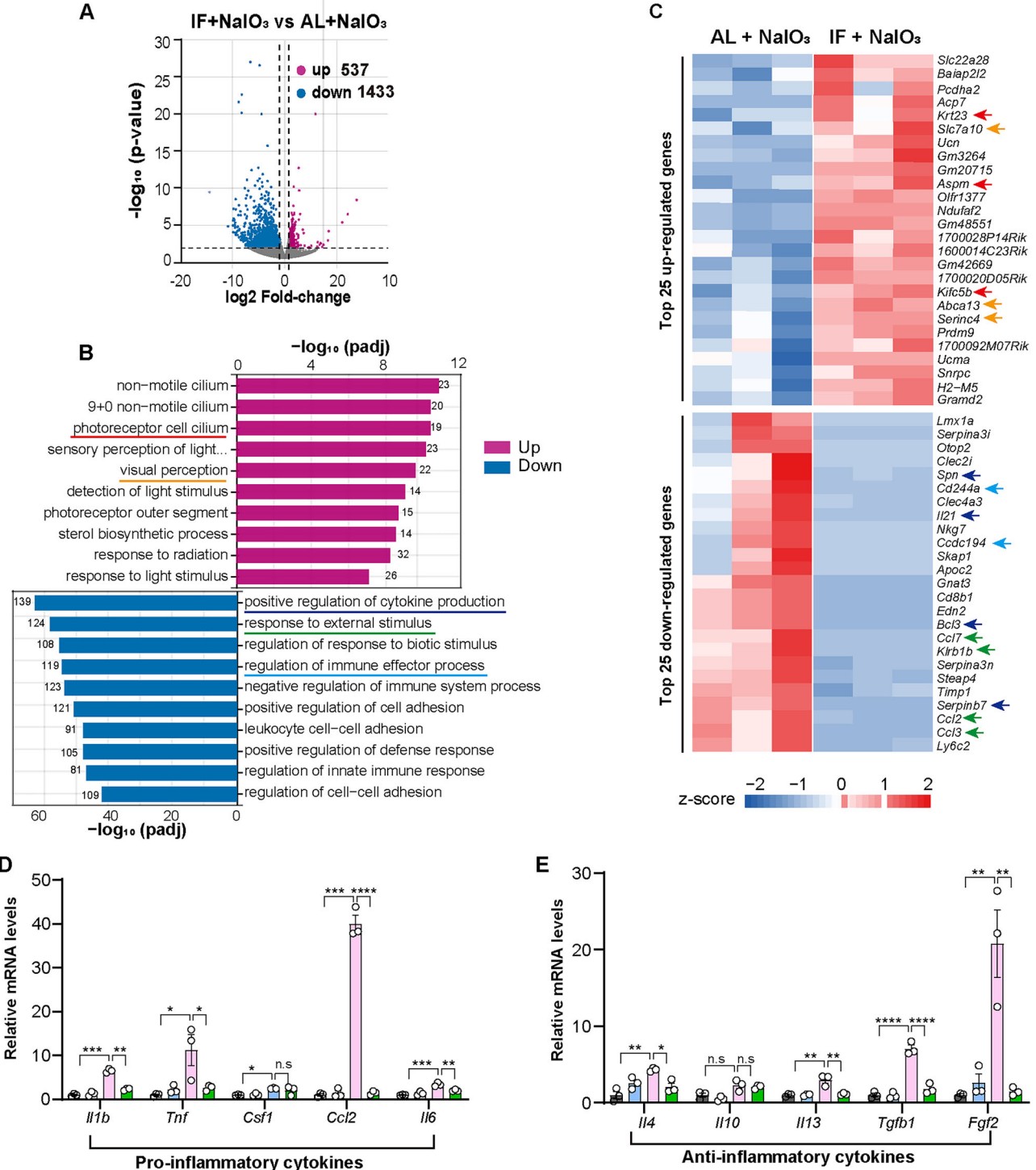

**Fig. 5 | Transcriptomic changes in the retinas of PBS and NaIO₃ exposed mice treated with AL or IF.** **A** Volcano plot of DEGs between the IF+NaIO₃ and AL+NaIO₃ groups; red dots indicate up-regulated genes, blue dots indicate down-regulated genes. **B** Gene Ontology (GO) functional enrichment analysis showing significant biological processes up- or down-regulated in IF+NaIO₃ compared to AL+NaIO₃. Red and blue lines highlight the pathways associated with photoreceptors and inflammation, respectively. **C** Heatmap exhibits the top up-regulated and down-regulated genes; red arrows indicate genes involved in photoreceptor function and blue arrows indicate genes associated with inflammation. **D** The mRNA levels of pro-inflammatory genes in each group at 7-days post NaIO₃ administration. *Gapdh*, *Actb*, *Ubc* were used as reference genes, $n = 3$ mice/group. **E** The mRNA levels of anti-inflammatory genes in each of the groups at 7-days post NaIO₃ administration. *Gapdh*, *Actb*, *Ubc* were used as reference genes, $n = 3$ mice/group. Data are expressed as the mean ± SD, *P* values were calculated using Two-way ANOVA with Tukey HSD post hoc tests in **D**, **E**. ****$P < 0.0001$, ***$P < 0.001$, **$P < 0.01$, *$P < 0.05$.

were decreased within the same regions in the IF group of mice (Fig. 6A–D). Moreover, immunostaining analysis of retinal and RPE flat mounts showed increased number and hypertrophic transformation, characterized by enlarged soma and thickened processes typical of activated/pro-inflammatory phenotypes, of microglia in NaIO₃-treated mice compared to AL + PBS controls, which were significantly attenuated by IF treatment (Fig. 6E–H). In AL+NaIO₃ retinas, the GFAP signals spanned nearly the entire retina, extending beyond the ganglion

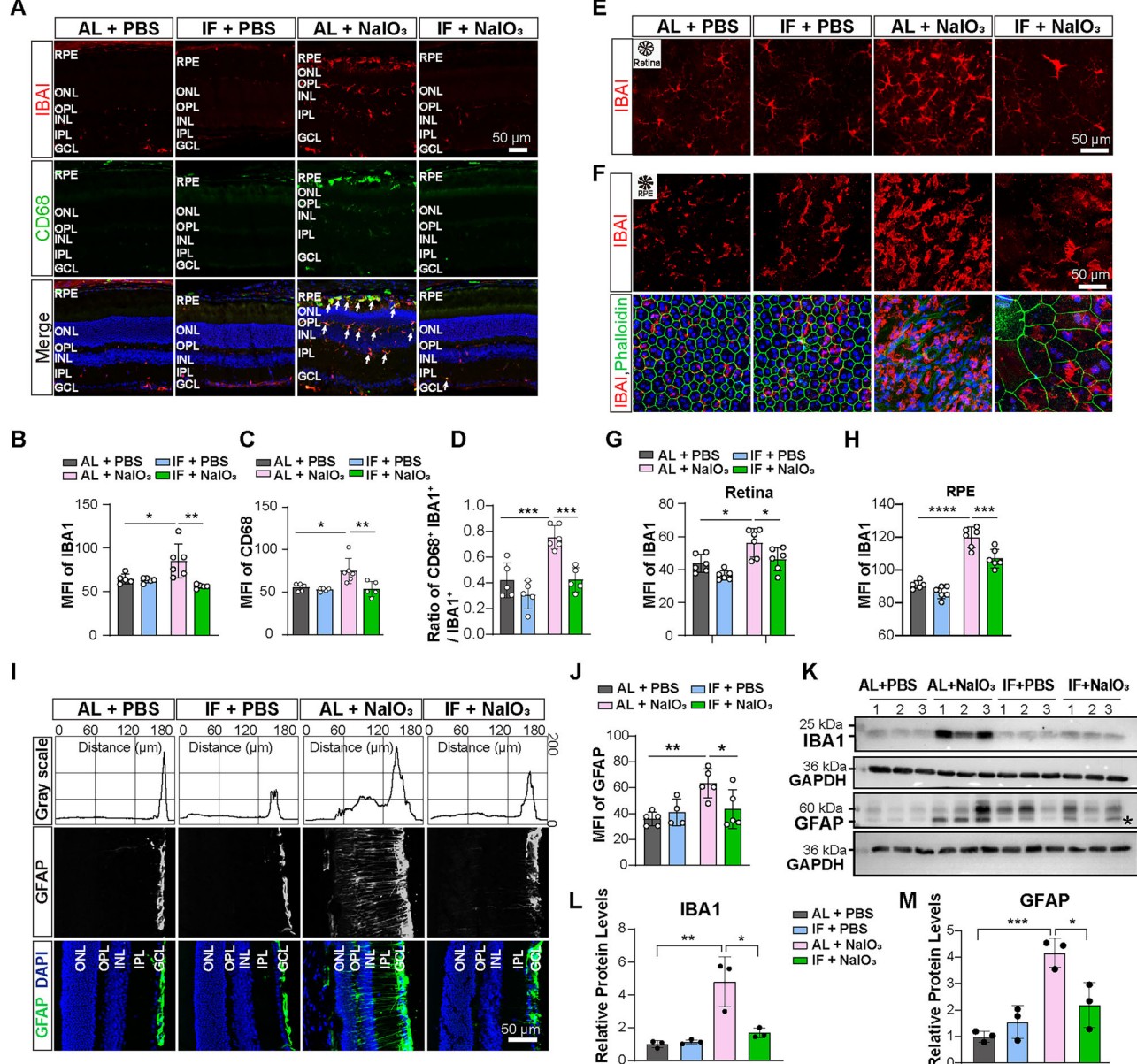

**Fig. 6 | IF treatment suppresses NaIO₃-inuced glial hyperactivation.**
**A** Representative images of IBA1 and CD68 immunostaining in retinal sections across different groups. Scale bar, 50 μm. **B–D** Mean fluorescence intensity of the IBA1 and CD68 signals and IBA1 + CD68 + /IBA1+ ratio. $n = 5$–6 mice/group. **E** Representative images of IBA1 immunostaining in retinal whole-mount flats across the groups. Scale bar, 50 μm. **G** Mean fluorescence intensity of the IBA1 signal in retinal whole-mount flats. $n = 6$ mice/group. **F** Representative images of IBA1 immunostaining in RPE whole-mount flats across the groups. Scale bar, 50 μm. **H** Mean fluorescence intensity of IBA1 signal in whole-mount RPE flats. $n = 6$ mice/group. **I, J** GFAP labeling of Muller glial cells in retinal sections. $n = 4$–5 mice/group. Scale bar, 50 μm. **K–M** Protein levels of IBA1 and GFAP as determined by Western blotting analysis. $n = 3$ mice/group. The asterisk in (**K**) indicates target bands. Data are expressed as the mean ± SD, P values were calculated using Two-way ANOVA with Tukey HSD post hoc tests in **B, D, F, H, J, K**. ****$P < 0.001$, ***$P < 0.001$, **$P < 0.01$, *$P < 0.05$.

cell layer (GCL) and nerve fiber layers. In contrast, IF-treated retinas exhibited GFAP signals confined to the GCL and nerve fiber layers, resembling the AL + PBS specimens (Fig. 6I, J). These results were confirmed by Western blotting, which showed that the protein levels of IBA1 and GFAP in the AL group were significantly upregulated by NaIO₃ and significantly suppressed by IF treatment (Fig. 6K–M). These results indicate that IF treatment suppresses the hyperactivation of glial cells in NaIO₃-induced retinal degeneration model.

### IF exerts mild retinoprotective effects in middle-aged and elderly mice
Recent studies have indicated that caloric restriction in later life can reduce age-associated frailty and reverse the alterations linked to the

aging process[39,40]. Therefore, we investigated whether IF treatment confers protective benefits against NaIO₃-induced RPE and retinal degeneration in older mice. First, we performed experiments using middle-aged mice, with mice initiating the IF regimen at 9 months of age, and were assayed at 12 months of age. Co-staining of RPE flat mounts with phalloidin and IBA1 revealed significantly less structural distortion in the middle and peripheral regions, as well as reduced IBA1 signals in the peripheral region of the RPE layer in IF + NaIO₃ mice than in AL + NaIO₃ mice (Fig. S8A–D). We subsequently extended the analyses to 16-month-old mice, finding that NaIO₃ caused more severe damage than younger mice, exemplified by almost complete loss of the hexagonal structure of the PRE cells, even in the middle and peripheral regions of the RPE layer (Fig. 7A). Even in mice

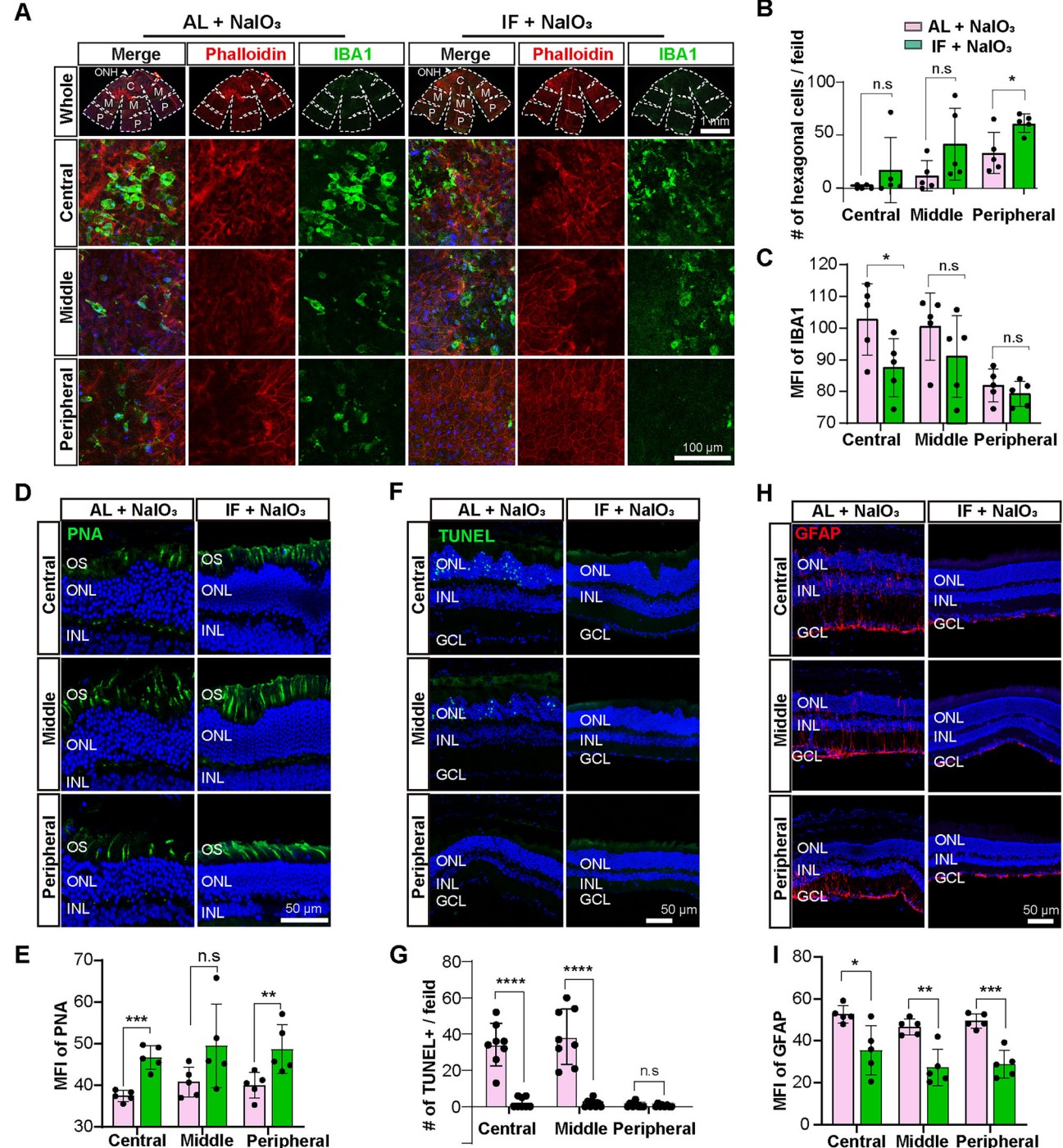

**Fig. 7 | IF exerts retinoprotective effects in older mice. A** Dual labeling of phalloidin and IBA1 showing the RPE integrity and microglial distribution in the RPE of 16-month-old mice. Scale bar, 1 mm (top row) or 100 μm (bottom three rows). **B, C** Quantification of hexagonal cells and IBA1 signal intensity in the RPE of 16-month-old mice. $n = 5$ mice/group. **D, E** Cone photoreceptors were identified by PNA labeling, with quantification of PNA mean fluorescence intensity across retinal sections. $n = 5$ mice/group. Scale bar, 50 μm. **F, G** Apoptotic cells were identified by TUNEL labeling and quantification of TUNEL fluorescence intensity across retinal sections. $n = 4$ mice/group. The left and right hemiretinas of the four mice were quantified independently in the figure. Scale bar, 50 μm. **H, I** Muller glial cell activation detected by GFAP immunostaining and quantification of GFAP mean fluorescence intensity in retinal sections. Scale bar, 50 μm. $n = 5$ mice/group. Data are expressed as the mean ± SD, $P$ values were calculated using unpaired two-tailed t-test in **B, C, E, G, I**. ****$P < 0.0001$, ***$P < 0.001$, **$P < 0.01$, *$P < 0.05$.

with this more severe phenotype, IF intervention partially alleviated the structural distortion of RPE cells (Fig. 7A–C). Further experiments labeling photoreceptors, apoptotic cells, and Muller cells in retinal sections from 16-month-old mice were conducted using PNA, TUNEL and GFAP staining, respectively. The analyses indicated that IF significantly inhibited NaIO₃-induced photoreceptor degeneration, cell death, and hyperactivation of Muller cells, albeit to a lesser degree, presumably due to the more severe damage or older age of the mice. (Fig. 7D–I, Fig. S9A, B). These results demonstrate that dietary restriction initiated in the later stages of life still exerts partial protective effects against RPE and retinal degeneration NaIO₃-induced retinal degeneration model.

## Discussion

Mounting evidence has indicated that IF exerts multiple beneficial effects on life span and various human diseases, including cancer, neurological disorders and metabolic dysfunctions[41–44]. However, its efficacy in combating retinal degeneration, particularly AMD, remains unclear. Our current investigation revealed that IF displayed robust protective effects in an oxidative stress-induced AMD model, attenuating ROS production, glial hyperactivation, and deficits in the retina and vision. Notably, this protective effect persisted even when IF was introduced in the middle and later life stages. Overall, our study provides novel insights into the beneficial effects of dietary restriction on enhanced retinal oxidative resistance, laying a foundation for new potential therapeutic strategies for AMD management.

The retina is one of the most energy-intensive organs in the body. Oxidative stress plays an important role in the pathogenesis of neurodegenerative diseases. For example, in AMD, excess free radicals attack the photoreceptor cells, triggering cell death. Consequently, the inability of the RPE to deplete free radicals leads to the degeneration of the RPE, and further induces photoreceptor loss[15,45]. In this study, we used a $NaIO_3$-induced dry AMD mouse model. Systemic $NaIO_3$ challenge recapitulates many of the key features of dry AMD, including oxidative stress, inflammation, and RPE dysfunction. Therefore, $NaIO_3$ has been widely used to study the pathogenesis and therapeutic effects of dry AMD[46–48]. Based on the preliminary experimental results, we chose a dosage of 25 mg/kg $NaIO_3$, which induced significant degeneration (greater than 50%) in the RPE layer and neuroretina. It is worth noting that in the $NaIO_3$-induced AMD model, the most severe damage to the RPE and retina occurred around the central area, with the damage diminishing with increasing distance from the central region. This pattern is consistent with previous reports[49,50]. Our data reveal age-related attenuation of IF-mediated protection against $NaIO_3$-induced retinal damage. This may reflect either diminished IF efficacy in aged mice or their heightened vulnerability to $NaIO_3$ toxicity, as evidenced by exacerbated midde and periphery RPE degeneration (Fig. S4, Fig.7A–C). While oxidative stress pathways likely contribute to this disparity, mechanistic distinctions between age-specific IF responsiveness and $NaIO_3$ sensitivity remain unresolved. Targeted investigations comparing molecular responses to IF and toxin challenge across age cohorts will clarify these interacting factors. Furthermore, the $NaIO_3$-induced AMD model is considered to be a more advanced model of dry AMD. It is worth studying whether IF protects the retina in other models of dry AMD, including light-induced, genetic, and wet AMD models.

In this study, a rapid decrease in body weight was observed in the initial week of IF treatment in mice. From the fourth week onwards, the mice began to regain weight. Later, they were able to maintain almost the same growth rate as the control group (Fig. 1A). However, by the end of the study, their body weights did not match those of the control group (approximately 3 g lighter). These changes in body weight are consistent with those reported in previous studies[27,51,52]. Over time, mice in the IF group gradually adapted and gained stable weight.

In the present study, we demonstrated that IF mitigates retinal oxidative damage induced by $NaIO_3$. Mechanistically, our findings indicate that IF can resist oxidative stress-induced damage to the RPE and retina by mitigating ROS production and the overactivation of microglia and Muller cells. However, the molecular pathways through which IF modulates glial cell activity remain unclear. We speculated that the gut microbiota may serve as this missing link, as IF can alter the composition of the gut microbiota, leading to an environment that favors anti-inflammatory SCFAs, such as butyric acid and propionic acid[53,54]. This change can have systemic effects as the gut microbiota interacts with the immune system. However, it is also possible that autophagy is involved in this process. Indeed, research has shown that IF enhances autophagy, defined as the process by which cells degrade and recycle their own components[55,56]. Autophagy can remove pro-inflammatory cellular debris and contribute to the reduction of cell death. Sirtuin activation may mediate the effects of IF on AMD. Sirtuins are a family of proteins that regulate cellular health and have been shown to exert anti-inflammatory properties. IF has further been

shown to activates sirtuins, which may reduce inflammation[44,57,58]. Although our study sheds light on the anti-inflammatory effects of IF in the retina, the pathways through which IF modulates the inflammatory response in our AMD model remain to be delineated. Future studies should aim to uncover specific protective mechanisms and identify key molecular targets.

Recent studies have shown that initiating IF therapy in late life decreases age-related frailty and improves aging-induced neurological deficits. Our study was therefore extended to assess the efficacy of IF in mitigating retinal RPE damage and photoreceptor degeneration in mice aged 12 and 16 months. The data revealed that both middle-aged and elderly mice exhibited significantly greater RPE damage and photoreceptor cell degeneration following $NaIO_3$ administration than their younger counterparts. This observation indicates increased susceptibility to $NaIO_3$-induced oxidative stress with advancing age. Furthermore, the protective effects of IF were observed to be less substantial in middle-aged (12-month-old) mice than in young mice, and even less so in elderly mice (16-month-old). Although IF treatment provided near-complete protection against RPE damage and cell loss in young mice, it conferred only partial protection in middle-aged and elderly mice. This variation may be due to the relatively elevated basal oxidative stress within the retina of aged mice, which makes them more sensitive to external stimuli. Consequently, they experienced more severe tissue damage under equivalent stimulus conditions, which partially obscured the original protective effects of IF. To validate this hypothesis, the protective effect of IF in older individuals was further assessed by reducing sodium iodate concentration, thereby mitigating the degree of retinal damage in these individuals.

Certain limitations should be considered when interpreting the outcomes of our study. Firstly, we used only male C57BL/6j male. Although no significant sex heterogeneity has been reported in the incidence of AMD, the inclusion of female subjects in future studies will be imperative to ensure a comprehensive understanding of the effects of IF interventions. Secondly, While the $NaIO_3$ model recapitulates acute oxidative injury rather than chronic AMD progression, its relevance persists through shared pathophysiology: RPE oxidative damage represents a hallmark of early AMD. Our findings elucidate dietary modulation of retinal stress resilience - a mechanism potentially applicable across injury timescales. Subsequent investigations employing aging models or multifactorial AMD systems will help elucidate therapeutic scope of IF intervention. Additionally, the current IF regimen was implemented as a preconditioning prior to $NaIO_3$ injury, which does not recapitulate lifelong dietary patterns or therapeutic interventions initiated after AMD onset. Future studies should explore whether IF commenced post-diagnosis in chronic models (e.g., $ApoE^{4/-}$ mice with drusen-like deposits) can halt established degeneration. Finally, it would be intriguing to examine whether regular intermittent fasting in early life exerts long-term beneficial effects on inflammatory immunity or antioxidant capacity in later life, thereby distinguishing between the acute protective effects observed in this study and lifelong metabolic adaptations.

In conclusion, our data showed that IF treatment reduced oxidative damage-induced RPE fragmentation, delayed the migration of microglia and Muller cells to the RPE and outer neural retina, and alleviated photoreceptor degeneration. These findings indicate this treatment as a novel therapeutic strategy for treating dry AMD and other retinal injuries caused by oxidative stress.

## Materials and methods
### Animals
Eight-week-old male C57BL/6 J mice (body weight: 22–24 g) were sourced from Cavens Laboratory Animal Co., Ltd. (Changzhou, China). Middle-aged (9-month-old) and aged (13-month-old) male C57BL/6 J mice were obtained from the Animal Center of Xuzhou Medical University. All mice were caged under specific pathogen-free (SPF) conditions in temperature-controlled facilities maintained at 22–25 °C with a 12-h light/dark cycle. Animals were monitored daily during routine husbandry checks and once daily during experimental procedures. Any deviations from baseline health were recorded, and animals meeting predefined humane endpoints (e.g.,

irreversible decline in health status) were euthanized immediately under isoflurane anesthesia. A total of approximately 150 mice were used in this study. We have complied with all relevant ethical regulations for animal use. All animal-related experimental procedures were conducted in accordance with the ARVO Statement for the Use of Animals in Ophthalmic and Vision Research and were approved by the Animal Ethics Committee of Xuzhou Medical University (licence numbers: 202306T029).

## Intermittent fasting

Body weight-matched male mice were randomly assigned to one of two dietary regimens: ad libitum (AL) or alternate-day fasting (IF) group. During the 12-week dietary regimens, Mice were maintained under a 12:12 h light/dark cycle, with lights on from 7:00 AM to 7:00 PM. Food was removed at 5:30 PM (1.5 h before the dark phase onset) for 24 h fasting (24 h of fasting) and provided at the same time on the following day (24 h of feeding) with ad libitum access; while AL group were allowed unrestricted access to food and water. Food consumption and body weight of both groups were recorded on a weekly basis. All treatment and measurement order was randomized between groups using block randomization with a 1:1 allocation ratio to eliminate temporal bias.

## Blood glucose test

The concentrations of blood glucose were quantified from tail vein blood samples of mice following a 6-h fasting period (9:00-15:00), using a Bayer handheld glucometer (ASCENSIA, China).

## Ketone determination

A ketone body detection kit (Solarbio, Beijing, China) was used to determine blood ketone levels in mouse blood samples following the manufacturer's instructions. Briefly, blood samples were collected from anesthetized mice and were centrifuged to separate the serum. 20 µL of the serum was mixed with 980 µL of reaction solution. After incubation for 20 s, the absorbance was measured using a spectrophotometer (PERSEE, China). Standard curve was included to determine ketone concentrations in samples. ketone body detection kit and other normal reagents have been listed in Supplementary Table S1.

## NaIO$_3$-induced mouse model of AMD

To establish an AMD model, mice were intraperitoneally injected once with sodium iodate (MilliporeSigma, MA, USA). 0.2% NaIO$_3$ solution was freshly prepared in PBS, sterilized through a 0.22 µm filter membrane, and injected intraperitoneally to mice. Injections of 10, 25, 30, 40 mg/kg were tested in pilot experiments to determine the optimal concentration, and a dosage of 25 mg/kg was selected for experiments. The control group received an equal volume of sterile PBS. The diet regimes (AL or IF) were maintained after NaIO$_3$ injection until end of experiments. Tissue samples were harvested 7 days post injection for molecular and biochemical assays, or 3 weeks post-injection for histopathological analyses. Sample sizes (n values) are provided in each figure legend and were determined based on established literature reporting similar disease models[59]. No animals were excluded. Data exclusion criteria during analysis were predefined as values outside the 1.5× interquartile range (IQR) of the group median. For example, in the IF+NaIO$_3$ group of Fig. 2H, one data point was excluded due to exceeding the predefined IQR threshold.

## Optomotor response (OMR) test

The optokinetic motor response (OMR) was assessed to evaluate visual function. This assessment was performed in accordance with a previously established protocol[60,61]. Briefly, the OMR device comprises a square frame made up of four equally sized computer screens, with reflective mirrors placed on the top and bottom surfaces. A platform was located in the center of the bottom mirror to hold the mouse, and a camera was positioned in the center opening of the top mirror (Fig. 1I). The sinusoidal wave grating animation used for the test was generated using MATLAB. Before starting the experiment, the mice were placed in a dark light environment for 24 h or

10 min, which were respectively used for the dark and light adaptation tests. Once the experiment was initiated, the mice were placed on the mouse platform and allowed to acclimatize to the surrounding environment for 2 min. Subsequently, the grating animation was played, and video recording commenced. During the test, if a mouse fell off the platform, it was returned to the platform to continue with the experiment. After the experiment, the recorded videos were analyzed to determine the number of head swings by each mouse at each stage. The counted number of head swings per stage indicated the contrast sensitivity, while the grating density with the most head swings indicated the visual acuity. The responses were recorded based on a blindness paradigm.

## Whole mount RPE preparation

Following euthanasia, eyeballs were enucleated and rinsed with pre-chilled PBS. The excess connective tissue on the posterior side of the eyeballs was gently excised under a dissecting microscope. The ocular specimens were fixed in freshly prepared 4% paraformaldehyde (PFA) for 5 min at room temperature. Subsequently, the cornea, lens, and vitreous were removed and the remaining tissues were left in PB-salt solution (140 mM NaCl and 2.7 mM KCl) for 20 min to facilitate the detachment of the retina from the eyecup. The eye cups were fixed in PFA for 1 h and examined under a dissecting microscope to peel off the retina. The other portion, the RPE-choroidal complex, was sectioned into petal shapes for the experiments.

## RPE flat labeling and degeneration analysis

For immunostaining of mouse RPE, flat mounts were placed in a decolorizing solution (3% hydrogen peroxide and 1% potassium hydroxide) for 10–15 min to eliminate melanin. Lipid droplet was stained with Nile red (0.5 µg/mL; MKBio) for 5 min followed by immunofluorescence. After rinsing once in PBS, the samples were incubated in blocking buffer (PBS with 0.3% Triton X-100, 1% BSA, 10% normal goat serum) for 1 h, followed by incubation with primary antibodies overnight at 4 °C. The samples were then washed three times followed by a 2-h incubation with secondary antibodies. The antibodies used for immunostaining are listed in Supplementary Table S2. After washing five times and countstaining with DAPI, the samples were mounted on slides for image acquisition using a laser scanning confocal microscope (Leica, Wetzlar, Germany). The entire RPE flat mount was segmented into three distinct regions extending from the optic nerve head (ONH) towards the edge (central: <1 mm, mid-peripheral: 1–2 mm, peripheral: >2 mm from the optic nerve head) for imaging, with one field per quadrant of the flat mounts. Images were analyzed using the ImageJ software (version 2.0). Numbers of RPEs were quantified by counting the hexagonal cells revealed by F-actin, Phalloidin, or TJP1. POS and lipid droplet were quantified by Mean fluorescent intensity (MFI) of PNA and Nile red. For quantification of mean fluorescent intensity (MFI), ImageJ software was utilized. Background subtraction was performed, and the fluorescence threshold was established based on negative control samples. Macro processing was then employed to automatically calculate the MFI values.

For pigment aggregation analysis, bright-field images of RPE flat mounts were acquired by optical microscope (Nikon Eclipse E100, 4, 20× objective). Pixel intensity of the RPE pigment was measured using ImageJ, with values normalized to the AL + PBS group. Pigment aggregation was presented as coefficient of variation (SD/mean of the pixel intensity).

## Retina processing and immunostaining

Enucleated eyes were fixed for 2 h in 4% paraformaldehyde in PBS at ambient temperature and sequentially infused with 30% sucrose and Tissue-Tek O.C.T Compound (Leica, Wetzlar, Germany). Cryosections of 12 µm thickness were generated using a cryostat (Leica, Wetzlar, Germany). Sections were sequentially stained with Hematoxylin and Eosin solutions (H&E) in accordance with the manufacturer's protocols. Images were captured using a BX53 microscope (Olympus, Tokyo, Japan). For immunostaining, retinal sections were rinsed in PBS, and permeabilized with 0.3% Triton X-100 for 10 min at 37 °C. Tissues were later blocked in a solution

containing 0.3% Triton X-100, 1% BSA, and 10% goat serum in PBS for an hour at 37 °C. The remaining steps were consistent with the established immunostaining protocol for RPE flat mount samples. Fluorescence signals of retinal sections were acquired at 20× (TUNEL, GFAP, NOX2, γH2AX) or 40× (PNA, RHO, cleaved CASP3, ROS, bright-field) magnification. Images were captured bilaterally in superior/inferior regions relative to the ONH to reduce positional bias.

The total number of cells within the retinal nuclear layers were counted using DAPI staining. TUNEL$^+$, γH2AX$^+$, and cleaved caspase-3$^+$ cells were quantified by counting the number of specific positive cells. Mean Fluorescence Intensity (MFI) measurements for GFAP, IBA1, CD68, ROS, and NOX2 were obtained across the entire retinal cross-section, using a method analogous to the RPE fluorescence signal quantification described previously. Quantification of PNA and RHO signals was confined to the region spanning from the outer nuclear layer (ONL) to the retinal pigment epithelium (RPE) layer within a defined Region of Interest (ROI).To evaluate the ONL and INL thickness, semicircular retinal contour was straightened for processing by ImageJ and divided into 18 equal-distance areas (9 inferior and 9 superior), at 250-μm intervals starting at the optic nerve head. Due to the nonuniform distribution of the nuclear layer, subsequent measurements of nuclear layer thickness were done manually in ImageJ.

### Reactive oxygen species (ROS) determination
Retina ROS levels were revealed by the CM-H2DCFDA dye. Fresh-frozen retinal sections were air-dried, rinsed with PBS, and incubated with a CM-H2DCFDA solution (Thermo Fisher Scientific, MA, USA; diluted 1:200 in PBS) for 30 min at 37 °C. After the incubation, sections were washed with PBS and fixed with 2% PFA for 10 min at ambient temperature. The fixed sections were counterstained with DAPI. Images were acquired using a fluorescent microscope (Leica, Wetzlar, Germany).

### TUNEL assay
Cryosections were allowed to dry completely before being washed with PBS. Proteinase K solution (20 ug/ml) was applied to treat the cryosections at 37 °C for 10 min and digestion was terminated by washing with PBS twice. Then apoptotic cells in the sections were labeled using a TUNEL assay kit (Vazyme, A112) following the manufacturer's manual. Briefly, the sections were incubated in TUNEL equilibration buffer for 15 min and then incubated in TUNEL reaction buffer for 2 h at 37 °C. Sections were washed three times for 5 min each with PBS containing 0.2% Tween-20. Sections were counterstained with DAPI. The fluorescence signals of dUTP-fluoresceine and DAPI were imaged with fluorescence microscope (Leica, Wetzlar, Germany).

### Western blotting
The eyes were enucleated after euthanasia and immediately dissected to remove anterior segment. Both eye cups from the same mouse were placed in a tube with RIPA buffer (Beyotime, Shanghai, China), supplemented with 1% protease inhibitor cocktail (Beyotime, Shanghai, China) and 5 mM EDTA. After sonication and centrifugation, protein content was quantified in each lysate using the BCA Protein Assay Kit (Beyotime, Shanghai, China). Western blotting was performed as previously described[62]. Protein levels were quantified using the ImageJ software (version 2.0). Antibodies used in this assay were listed in Supplementary Table S2.

### RNA extraction and quantitative polymerase chain reaction (qPCR)
Both RT-PCR and qPCR were performed following the manufacturer's manuals. Briefly, both eye cups from the same mouse were collected and lysed in 1 mL of TRIzol reagent (Thermo Fisher Scientific, MA, USA). Tissues were homogenized using a handheld grinder for 30 s, and 0.2 mL of chloroform was added. After centrifugation, the aqueous phase was collected, and 0.5 mL of isopropanol was added. The RNA was precipitated in -20 °C overnight with addition of 5 μg glycogen. After centrifugation, the RNA was washed using 75% ethanol and dissolved in 20 μL of RNase-free water. RNA was quantified using Nanodrop 2000 (Thermo Fisher Scientific, MA, USA). cDNAs were synthesized from the total RNA using the ABScript II One Step SYBR Green RT-qPCR Kit (ABclonal, Hubei, China). qPCR analyses were performed using the Genius 2× SYBR Green Fast qPCR Mix (ABclonal, Hubei, China) in a StepOne Real-Time PCR System (Thermo Fisher Scientific, MA, USA). Relative expression levels were calculated using the ΔΔCt method normalized to the geometric mean of three reference genes (*Gapdh, Actb, Ubc*) to avoid possible errors introduced by normalization with single reference gene[63]. The sequences of all primers used in this study are listed in Supplementary Table S3.

### RNA-seq analysis
RNA was isolated from TRIzol-lysed fresh eye cup tissues, ensuring high integrity (RIN > 8) and purity (260/280 > 1.9; 260/230 > 1.8). Library preparation and sequencing were performed by APExBIO Technology (TX, USA). Briefly, 500 ng of total RNA was poly-A selected using beads, fragmented by metal-ion hydrolysis, and converted into double stranded cDNA. The library quality was validated, followed by single-end 150-bp sequencing on a PE150 platform. Data quality was verified, and reads were mapped to the GRCm39 mouse genome. Reads Per Kilobase Million (RPKM) normalization accounted for sequencing depth and gene length. DESeq2 analysis discerned differential gene expression between IF+NaIO$_3$ and AL +NaIO$_3$ samples, with differentially expressed genes defined by thresholds of FDR < 0.05 and fold-change ≥ 1.5. Gene Ontology analysis was performed using the R package gprofiler2. RNA-seq data in this study have been uploaded to the GEO repository under accession number: GSE271791.

### Statistical analysis
Sample sizes for all experimental groups are explicitly documented in respective figure legends. All multi-group comparisons were performed using two-way ANOVA with independent factors of IF treatment and NaIO$_3$ challenge. Post hoc comparisons between specified groups were conducted only when main effects/interactions reached significance (Alpha=0.05), ensuring stringent control of Type I errors. Unless indicated otherwise, data are presented as the mean ± SD. Statistical significance is denoted as *$P < 0.05$, **$P < 0.01$, ***$P < 0.001$.

### Reporting summary
Further information on research design is available in the Nature Portfolio Reporting Summary linked to this article.

### Data availability
The RNA-seq data in this study have been uploaded to GEO under accession number: GSE271791. The raw data supporting the statistical figures are provided in Supplementary Data 1. Raw Ct values from qPCR experiments and datasets used for statistical analysis are included in Supplementary Data 2. Uncropped Western blot images have been included in the Supplementary Information (Figure S10). Other data will be made available upon request.

### Code availability
The sinusoidal wave grating animation used for the OMR test was generated using MATLAB, The custom code used in this study has been deposited in the GitHub repository (https://github.com/DorJingzhen/OMR-Optical-Grating.git) and archived via Zenodo (https://doi.org/10.5281/zenodo.16259292)[64]. This version of the code is freely available without restrictions.

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

## Acknowledgements

This study was supported by grants from the National Natural Science Foundation of China (82371401, 32100769, 32300485), Jiangsu Provincial Department of Education (JSSCBS20221563, 22KJB180028, 24KJA310011), and Xuzhou Medical University (D2020054, D2021025, JBGS202202, KYCX24_3100).

## Author contributions

**Jingzhen Li:** Writing – original draft, Conceptualization, Data curation, Formal analysis, Funding acquisition, Project administration, Software. **Beibei Wang:** Data curation, Formal analysis, Investigation, Methodology. **Pinjie Liu**: Data curation. **Xuecheng Qiu:** Conceptualization, Methodology. **Qiyun Bian:** Investigation, Methodology. **Congxin Shen:** Visualization, Resources. **Yanyan Li:** Resources. **Mengwen Shao:** Resources. **Meng Li:** Writing – review & editing, Funding acquisition, Conceptualization, Supervision, Project administration.

## Competing interests

The authors declare no competing interests.
