## [Transparent Peer Review file · Communications Biology]

Intermittent fasting attenuates glial hyperactivation and photoreceptor degeneration in a NaIO₃-induced mouse model of age-related macular degeneration

Corresponding Author: Professor Meng Li

Version 0:

Reviewer comments:

Reviewer #1

(Remarks to the Author)

In this study the authors aim to understand whether intermittent fasting (IF) alleviates retinal degeneration and neuroinflammation in the NaIO₃ mouse model for age-related macular degeneration (AMD). While the topic is of interest, particularly given recent human observations linking IF to a decreased risk of AMD (PMID: 35809657), a major concern arises regarding the interpretation of the findings in light of the experimental setup. The current experimental design, with its inherent limitations in modelling AMD and the short time frame of the study, does not substantiate claims that IF alleviates retinal damage in AMD. Instead, the findings suggest that mice subjected to IF prior to intraperitoneal injection of NaIO₃ demonstrate increased resistance to retinal degeneration induced by this chemical method. This overstatement should be addressed throughout the manuscript, including in the title, abstract, introduction, and results, to accurately reflect the study's scope and findings. For example, statements such as "deficit in phagocytosis by RPE cells was rescued by IF treatment" (page 8 lines 20-21) should be avoided, as they imply conclusions that exceed the evidence provided. In addition, the discussion should be revised for the same reason, as the current experimental approach limits the interpretation of the findings regarding protection against AMD.

Major concerns

- As shown in all Figures, the use of t-tests for comparing multiple groups is not statically appropriate. Given the presence of multiple comparisons, a multivariate analysis, such as ANOVA or another appropriate statistical method followed by post hoc analysis, is required to ensure the validity and robustness of the results. This adjustment is crucial to account for potential type I errors and provide a more reliable interpretation of the data.
- Supplemental Fig S6c-d, Fig 4d and Fig 5d-e – please include housekeeping genes to which data was normalized. To ensure consistency of mRNA data, it would be essential to normalize to more than one house-keeping gene (PMID: 12184808)
- Middle-aged and elderly mice (Section 2.7): Control groups are lacking (i.e. non-NaIO₃ induced). This hinders the ability to interpret the severity of damage induced by NaIO₃ and protection conferred by intermittent fasting.
- Fig 1J-K optokinetic responses: A control group consisting of intermittent fasting + PBS should be included, as shown in the other figures, to establish baseline levels.
- The manuscript presents a significant amount of morphometric data; however, minimal methodological details are provided. Adding comprehensive descriptions of the morphometric analysis methods, including how measurements were obtained and quantified, will be essential to validate the data interpretations and ensure reproducibility.
- Page 20 – Following previous comment, how do the authors quantify microglial activation? A brief description or definition of the morphological characteristics of activated microglia would enhance clarity. The authors should consider using an alternative marker of microglial activation, such as CD68, as Iba1 expression alone may not be sufficient to confirm activation. Additionally, better representative images, particularly for the retina, should be provided to accurately depict the observed changes.
- For primary antibodies the host should be also added in the Supplemental material. For instance, GFAP antibody appears to be raised in mice, as non-specific IgG binding is detected in the retinal capillaries across all groups (Fig 6). I recommend quantifications using a different host (e.g. rabbit) to avoid noise.
- Page 14 – "These results suggest that IF treatment suppresses NaIO₃-induced RPE and retinal degeneration by reducing ROS production" This statement is not accurate. While a correlation may exist between IF treatment and reduced ROS

production, mechanistic studies are required to determine causality. Additionally, some of the data appear incongruent, as NOX2 seems primarily localized to the inner limiting membrane (ILM) rather than the outer retina, where the observed damage is more pronounced. Middle and peripheral regions do not exhibit a similar spatial pattern (Supplemental Fig 7). This discrepancy should be addressed with a different oxidative stress marker to ensure the spatial relevance of the findings.

Minor comments:

- Page 4 lines 7-8 “Recently, a population based association study found breakfast frequency per week as a risk factor for AMD”. Based on that study findings this should be rephrased given the potential misleading interpretation that increased frequency in breakfast is associated with AMD.
- Page 5, lines 10-11: To aid non-specialized readers, please provide a brief contextualization with previous literature explaining why the results in Figures 1B–1H indicate the successful establishment of the intermittent fasting (IF) intervention. For example, clarify the significance of increased ketone bodies, such as what elevated β -hydroxybutyrate (BOH) levels in the blood typically indicate regarding metabolic adaptation to fasting.
- Page 5 lines 10-11 – Choice of 25 mg/kg NaIO₃ could be further supported by previous literature (e.g. PMID: 38158173)
- Supplemental Fig S3b – please include layer annotations

Reviewer #2

(Remarks to the Author)

In this study, the author highlights the significance of intermittent fasting in mitigating RPE damage and preserving RPE morphology and function in the NaIO₃-induced retinal degeneration model, thereby reducing photoreceptor cell death by alleviating oxidative stress, glial activation, and neuroinflammation. The manuscript is well-written, and the results and conclusions are clearly presented. Here are some suggestions for the author's consideration:

1. Diet (normal vs. intermittent fasting) and disease model (PBS vs. NaIO₃) are two independent variables. Since the author is evaluating the effects of diet and disease model, as well as their interaction (i.e., the effect of intermittent fasting on the mouse retinal damage model), I would suggest using a two-way ANOVA instead of a t-test when comparing effects among AL+PBS, IF+PBS, AL+NaIO₃, and IF+NaIO₃.
2. For the RNA-seq data, the author may consider adding an interaction term (~ diet + disease model + diet:disease model) when performing DESeq2 differential expression analysis. This approach would provide a clearer understanding of the individual effects of intermittent fasting and NaIO₃ on retinal mRNA expression, as well as how different mice respond to NaIO₃ under two dietary conditions.
3. For the aging mouse study (Figure 7), it would be helpful to provide data on the thickness of the ONL between the AL+NaIO₃ and IF+NaIO₃ groups. Additionally, the GFAP staining did not show differences between the AL+NaIO₃ and IF+NaIO₃ groups in the middle region of the retina.
4. On page 2, line 23, the statement “Dry AMD, also known as geographic atrophy, accounts for approximately 90% of all the AMD cases”, should be revised for accuracy. While dry AMD accounts for about 90% of AMD cases, geographic atrophy specifically refers to the later or advanced stage of dry AMD.
5. On page 8, line 2, the statement “the AL+NaIO₃ group exhibited significant pigment fragmentation and accumulation within the RPE layer”, should be supported with quantitative data when using the term “significant.”
6. In Figure 5C, the label should be corrected to IF+NaIO₃.
7. On page 20, line 1, the statement “Immunostaining of retinal flat mounts revealed the morphological characteristics of ramified and activated microglia in the NaIO₃-challenged mice (Fig. 6B)”, please ensure whether the correct figure reference should be Figure 6C.
8. In Figure 6D, please ensure the y-axis label is correctly labeled as “IBA1+ intensity.”

Reviewer #3

(Remarks to the Author)

Version 1:

Reviewer comments:

Reviewer #1

(Remarks to the Author)

The authors have addressed most of the suggested revisions. However, I would like to raise a methodological concern regarding the quantification of fluorescence intensity. The use of integrated density as a readout is not ideal, as it is directly influenced by the size of the area measured—larger areas will inherently yield higher values, which may confound interpretation. I strongly recommend that the authors express the data as mean fluorescence intensity (MFI) after appropriate background subtraction in all panels involving fluorescence measures. This approach normalizes for area and will provide a more accurate representation of fluorescence signal, minimizing potential artifacts arising from variable region sizes.

Reviewer #2

(Remarks to the Author)

Thank you to the authors for providing the revised manuscript. Please find additional comments below:

1. In Figure 1J and Figure S2B, could the authors clarify the detailed statistical methods used for comparing the number of head rotations at different frequency among four groups?
2. Hprt1 shows a CT value around 28, suggesting relative low abundance in the samples. This makes it a less ideal choice for a house-keeping genes.
3. In Figure S7A, the PCA reveals that the variance captured by PC1 and PC2 is primarily driven by heterogeneity within AL+NaIO3 group. This within-group variability is also observed in Figure 5C. The authors should be cautious in interpreting global sample similarity. Additionally, the authors should check the sample-level QC (e.g., sample quality, sequencing quality, mapping quality) to rule out any technical artifacts that may contribute the presence of outlier sample.

Version 2:

Reviewer comments:

Reviewer #1

(Remarks to the Author)

As noted in my previous comment, the use of integrated density to quantify marker expression in retinal areas is not a reliable approach, even when background subtraction is applied. This method combines signal intensity with the size of the measured area, which is particularly problematic in the retina, where fluorescence intensity can vary substantially depending on the extent and anatomical region selected.

This represents a significant methodological limitation. I strongly recommend that the authors reanalyze their retinal data using mean fluorescence intensity (MFI), which provides a more accurate and standardized measure of expression per unit area. Doing so would greatly enhance the robustness, reproducibility, and interpretability of the results.

Reviewer #2

(Remarks to the Author)

The authors have addressed most of points raised during the review. I have no further questions.

Version 3:

Reviewer comments:

Reviewer #1

(Remarks to the Author)

The authors have successfully addressed all concerns

Reviewers' comments:

Reviewer #1 (Remarks to the Author):

In this study the authors aim to understand whether intermittent fasting (IF) alleviates retinal degeneration and neuroinflammation in the NaIO₃ mouse model for age-related macular degeneration (AMD). While the topic is of interest, particularly given recent human observations linking IF to a decreased risk of AMD (PMID: 35809657), a major concern arises regarding the interpretation of the findings in light of the experimental setup. The current experimental design, with its inherent limitations in modelling AMD and the short time frame of the study, does not substantiate claims that IF alleviates retinal damage in AMD. Instead, the findings suggest that mice subjected to IF prior to intraperitoneal injection of NaIO₃ demonstrate increased resistance to retinal degeneration induced by this chemical method. This overstatement should be addressed throughout the manuscript, including in the title, abstract, introduction, and results, to accurately reflect the study's scope and findings. For example, statements such as “deficit in phagocytosis by RPE cells was rescued by IF treatment” (page 8 lines 20-21) should be avoided, as they imply conclusions that exceed the evidence provided. In addition, the discussion should be revised for the same reason, as the current experimental approach limits the interpretation of the findings regarding protection against AMD.

Response: We understand the concerns of the reviewer about interpretation of the results. We have revised the text to be more consistent with data presentation and interpretation in the research field.

- 1) Although the NaIO₃ model is widely used for modeling AMD in various species of animals (PMID: 2747981), we agree with the reviewer that the model has its limitations as it represents an acute chemical injury paradigm rather than a comprehensive AMD model. We therefore have implemented comprehensive revisions to clearly specify our model as “NaIO₃-induced mouse model of AMD” to more accurately reflect the experimental design.
- 2) As for the IF diet regime, we made revisions to clearly indicate in the abstract, introduction, as well as schematic diagrams in Figures 1A and S8A, that the diet management was prior to NaIO₃ injection.
- 3) To our knowledge, claims like our “IF alleviates retinal damage” is well accepted in similar experimental design reported in high-profile publications (e.g., PMID: 35624490, 35732737). Moreover, other types of long-term intervention, such as exercise, prior to disease modeling were also reported with similar phrases (e.g., PMID: 37150437, “Exercise alleviates neovascular age-related macular degeneration by inhibiting AIM2 inflammasome in myeloid cells”). However, to address the reviewer’s concern about result interpretation, we made

revisions to clearly indicate our experimental designs and revised some conclusions to more precisely interpret the data and results.

Major concerns

- As shown in all Figures, the use of t-tests for comparing multiple groups is not statically appropriate. Given the presence of multiple comparisons, a multivariate analysis, such as ANOVA or another appropriate statistical method followed by post hoc analysis, is required to ensure the validity and robustness of the results. This adjustment is crucial to account for potential type I errors and provide a more reliable interpretation of the data.

Response: We agree with the reviewer that a multivariate analysis is more appropriate for our data. In accordance with the statistical recommendations, we have reanalyzed all multi-group comparisons (Figure 1J, K; Figure 2-6; Figure S2-S4; Figure S6; Figure S7) using two-way ANOVA with independent factors of diet (IF vs AL) and disease model (NaIO₃ vs PBS). Post hoc comparisons between specified groups were conducted only when main effects/interactions reached significance (Alpha=0.05), ensuring control of Type I errors. Corresponding updates have been integrated into the Methods section (Page 40, Section 4.15), figure legends, and graphical presentations.

- Supplemental Fig S6c-d, Fig 4d and Fig 5d-e – please include housekeeping genes to which data was normalized. To ensure consistency of mRNA data, it would be essential to normalize to more than one house-keeping gene (PMID: 12184808)

Response: In accordance with the recommendation, we have included two additional widely used housekeeping genes, *β-Actin* and *Hprt1*. Geometric mean of the three reference genes was used for normalization of qPCR data as recommended by the cited publication (PMID: 12184808). There are no changes in our conclusions using the revised normalization method, with mild changes in the quantification values. The normalization strategy has been added to the revised method section (Page 39, Section 4.13), citing Vandesompele *et al.* (PMID: 12184808) as methodological reference. Names of the three reference genes have been added to relevant figure legends. Additionally, raw Ct values and analyses of qPCR data have been compiled in **Supplementary Excel 2**.

- Middle-aged and elderly mice (Section 2.7): Control groups are lacking (i.e. non-NaIO₃ induced). This hinders the ability to interpret the severity of damage induced by NaIO₃ and protection conferred by intermittent fasting.

Response: Thank you for the insightful critique. Although we were unable to include the control groups in all the experiments due to limited resources, we added an AL+PBS control group in the new Fig. S9 measuring the ONL thickness. The results indicated marked ONL thinning in AL+NaIO₃ group and a mild increase in IF+NaIO₃ group compared with AL+NaIO₃ group. Similar to Fig. 7, the beneficial effects of IF were less profound in the older mice. Therefore, we added “mild” and “partial” to the revised section title and conclusions for more precise interpretation of our results.

- Fig 1J-K optokinetic responses: A control group consisting of intermittent fasting + PBS should be included, as shown in the other figures, to establish baseline levels.

Response: Thank you for your constructive feedback. We acknowledge the significance of including the IF + PBS control group. As advised, we have included data on the IF + PBS group in the revised Figure 1J-K and Figure S2B, C.

- The manuscript presents a significant amount of morphometric data; however, minimal methodological details are provided. Adding comprehensive descriptions of the morphometric analysis methods, including how measurements were obtained and quantified, will be essential to validate the data interpretations and ensure reproducibility.

Response: Thanks for your feedback regarding the methodological description of our morphometric analyses. We have added methodological descriptions on image acquisition, processing, and quantification of readouts from RPE and retinal flat-mount and cross section samples to sections 4.8 and 4.9 of the Methods.

- Page 20 – Following previous comment, how do the authors quantify microglial activation? A brief description or definition of the morphological characteristics of activated microglia would enhance clarity. The authors should consider using an alternative marker of microglial activation, such as CD68, as Iba1 expression alone may not be sufficient to confirm activation. Additionally, better representative images, particularly for the retina, should be provided to accurately depict the observed changes.

Response: Thank you for the critical comment. As the reviewer pointed out, although Iba1 can reveal the increase in number of microglia as one indicator of microglial activation, the pan-microglia marker Iba1 alone cannot demonstrate the activation state of microglia. Following the reviewer’s suggestion, we have included CD68 immunofluorescence to confirm activation of

microglia. Moreover, the morphology of microglia in the AL+NaIO₃ group shifted towards enlarged cell body and thick processes, characteristics of hypertrophic microglia (activated and pro-inflammatory). Representative high-resolution images and quantitative data have been updated in Figure 6A-D. The results showed that a significant increase in CD68⁺ microglia in NaIO₃-treated retinas ($p < 0.001$ vs. control), particularly in the outer nuclear layer (ONL) and subretinal space. Intermittent fasting (IF) reduced both Iba1⁺ and CD68⁺ signal intensity ($p < 0.001$), confirming its suppressive effect on microglial activation (Figure 6A-D). Additionally, we have replaced the retinal microglia images with higher-resolution examples that better illustrate intergroup morphological differences (Figure 6E).

- For primary antibodies the host should be also added in the Supplemental material. For instance, GFAP antibody appears to be raised in mice, as non-specific IgG binding is detected in the retinal capillaries across all groups (Fig 6). I recommend quantifications using a different host (e.g. rabbit) to avoid noise.

Response: Thank you for the insightful comments. The GFAP antibody was indeed raised in mice. The antibody was able to reveal changes in Müller cells. However, as the reviewer pointed out, noise raised by non-specific binding to the capillaries may interfere with accurate quantification to some extent. To eliminate non-specific IgG binding, we repeated the experiments using a rabbit-derived GFAP antibody (Proteintech, 16825-1-AP, Page22) and updated Figure 6I-J.

Information on host of primary antibodies has been added to **Supplemental Table S3 (Supplementary Information, Page 17)**.

- Page 14 – “These results suggest that IF treatment suppresses NaIO₃-induced RPE and retinal degeneration by reducing ROS production” This statement is not accurate. While a correlation may exist between IF treatment and reduced ROS production, mechanistic studies are required to determine causality. Additionally, some of the data appear incongruent, as NOX2 seems primarily localized to the inner limiting membrane (ILM) rather than the outer retina, where the observed damage is more pronounced. Middle and peripheral regions do not exhibit a similar spatial pattern (Supplemental Fig 7). This discrepancy should be addressed with a different oxidative stress marker to ensure the spatial relevance of the findings.

Response:

1) We agree that the original statement is not accurate. We have revised the conclusion as follows, "These results indicated that IF treatment reduced NaIO₃-induced ROS production and ameliorated oxidative damage to the retina." (Page 15, Lines 17-19).

2) We appreciate the reviewer's comment on the spatial inconsistency between NOX2 and retinal damage in the representative images. Following the reviewer's comment, we performed immunofluorescence of γ H2A.X (Page 16, Figure 4G), an indicator of oxidative DNA damage found in many pathological conditions including overproduction of ROS (PMID: 34073310). We also went through our original images of ROS, NOX2, and cleaved caspase 3. As shown in Figure R1 below, signals of ROS, γ H2A.X, and cleaved caspase 3 in the AL+NaIO₃ retinal sections were all found primarily around the nuclear layers, especially inside or around the ONL region. NOX2⁺ signals were also found around the ONL region. Together, the four signals around the ONL layer indicate ROS overproduction and oxidative damage in NaIO₃-treated retina. We apologize that some of the images in our last version of manuscript were not representative. We have replaced them with more representative images. We thank the reviewer for prompting this re-evaluation for better presentation of our data.

Figure R1. Fluorescence images of Cleaved caspase 3, CM-H2DCFDA, NOX2, and γ H2A.X from the AL+NaIO₃ retinal sections.

Minor comments:

- Page 4 lines 7-8 “Recently, a population based association study found breakfast frequency per week as a risk factor for AMD”. Based on that study findings this should be rephrased given the potential misleading interpretation that increased frequency in breakfast is associated with AMD.

Response: We acknowledge that the original statement could be misinterpreted as implying breakfast consumption itself increases AMD risk. To clarify, we have revised the text: “Recently, a population-based association study found that intermittent fasting by skipping breakfast correlates with a significantly reduced risk of AMD.” (Page 4, Lines 11-12).

- Page 5, lines 10-11: To aid non-specialized readers, please provide a brief contextualization with previous literature explaining why the results in Figures 1B–1H indicate the successful establishment of the intermittent fasting (IF) intervention. For example, clarify the significance

of increased ketone bodies, such as what elevated β -hydroxybutyrate (BOH) levels in the blood typically indicate regarding metabolic adaptation to fasting.

Response: Thank you for the thoughtful suggestion. We have added brief text with citation to provide background on physical and metabolic changes following IF (Page 5, Lines 10-17).

- Page 5 lines 10-11 – Choice of 25 mg/kg NaIO₃ could be further supported by previous literature (e.g. PMID: 38158173)

Response: Thank you for the helpful suggestion. The reference (PMID: 38158173) has been incorporated into the relevant section on Page 6, Lines 6-7 to provide further support for our choice of using 25 mg/kg NaIO₃.

- Supplemental Fig S3b – please include layer annotations

Response: Thank you for the suggestion. Annotations of the retinal layers have been added for better legibility (Supplemental Fig S3D, Page 4).

Reviewer #2 (Remarks to the Author):

In this study, the author highlights the significance of intermittent fasting in mitigating RPE damage and preserving RPE morphology and function in the NaIO₃-induced retinal degeneration model, thereby reducing photoreceptor cell death by alleviating oxidative stress, glial activation, and neuroinflammation. The manuscript is well-written, and the results and conclusions are clearly presented. Here are some suggestions for the author's consideration:

1. Diet (normal vs. intermittent fasting) and disease model (PBS vs. NaIO₃) are two independent variables. Since the author is evaluating the effects of diet and disease model, as well as their interaction (i.e., the effect of intermittent fasting on the mouse retinal damage model), I would suggest using a two-way ANOVA instead of a t-test when comparing effects among AL+PBS, IF+PBS, AL+NaIO₃, and IF+NaIO₃.

Response: We agree with the reviewer that two-way ANOVA is more appropriate for the multiple comparison of data. In accordance with the statistical recommendations, we have carefully reanalyzed all multi-group comparisons (Figure 1J, K; Figure 2-6; Figure S2-S4; Figure S6; Figure S7) using two-way ANOVA with independent factors of diet (IF vs AL) treatment and disease

model (NaIO₃ vs PBS) challenge. Post hoc comparisons between specified groups were conducted only when main effects/interactions reached significance (Alpha=0.05), ensuring control of Type I errors. Corresponding updates have been integrated into the Methods section (Page 40, Section 4.16), figure legends, and graphical presentations.

2. For the RNA-seq data, the author may consider adding an interaction term (~ diet + disease model + diet:disease model) when performing DESeq2 differential expression analysis. This approach would provide a clearer understanding of the individual effects of intermittent fasting and NaIO₃ on retinal mRNA expression, as well as how different mice respond to NaIO₃ under two dietary conditions.

Response: We agree with the reviewer that analysis with the interaction term provides the readers with a clearer picture of the effects of the main factors and their interaction. We therefore conducted additional bioinformatic analysis as follows.

- 1) PCA (principal component analysis) plot is widely used to provide straightforward visualization of overall transcriptomic differences between samples. As shown in the PCA plot below, in healthy (PBS-treated, triangles) conditions, transcriptomic profiles of IF mice (hollow triangle) were similar to those of AL mice (solid triangle), indicating a mild response of healthy mice to IF diet. In contrast, in the NaIO₃-induced disease model (square), mice on normal (AL) diet (solid square) presented marked deviation of gene expressions from healthy (PBS-treated) mice (triangles). In contrast, IF diet in disease model (NaIO₃-treated) mice (hollow square) was able to shift gene expressions back towards healthy mice (triangles). The result demonstrates that mice under AL and IF diet regimes responded to NaIO₃ differently. The figure has been included as Supplementary Figure S7A and interpretation of the plot has been added in the revised manuscript (Page 14, Lines 10-14).

Figure R2. PCA plot of RNA-seq samples from four groups of mice.

2) As suggested by the reviewer, we also performed DESeq2 analysis with a formula of “~ diet + disease_model + diet:disease_model”, and obtained gene lists significantly affected by each of the main factors (diet and disease model) and their interaction (Figure R3). Responses to NaIO₃ treatment in the AL mice were highly related to the main factors and the interaction, as 411 of 452 DEGs (AL_NaIO₃ vs AL_PBS) could be attributed to the factors (Figure R3, left panel). In contrast, only 10 of the 166 DEGs (IF_NaIO₃ vs IF_PBS) in the IF group of mice could be attributed to the factors (Figure R3, right panel).

Figure R3. Venn diagrams showing overlapping of genes differentially expressed under various conditions.

Together, the data indicates that mice on the AL diet were susceptible to NaIO₃-induced changes in gene expression profiles, while mice on the IF diet were more resistant to NaIO₃-induced AMD model.

3. For the aging mouse study (Figure 7), it would be helpful to provide data on the thickness of the ONL between the AL+NaIO₃ and IF+NaIO₃ groups. Additionally, the GFAP staining did not show differences between the AL+NaIO₃ and IF+NaIO₃ groups in the middle region of the retina.

Response: Thank you for these constructive suggestions.

1. We have performed statistical comparisons of the ONL thickness between the AL+PBS, AL+NaIO₃, and IF+NaIO₃ groups and added the results to the Supplementary Information (Supplementary Figure S9). This data supports the protective effect of intermittent fasting observed in the aging mouse model.

2. For the GFAP staining discrepancy, the GFAP antibody was derived from mouse, which might interfere with accurate quantification. We therefore repeated the experiment with a rabbit-derived GFAP antibody. The revised images and data have been updated in the revised manuscript (Page 25, Figure 7H-I). The new data showed a significant reduction of GFAP signals. However, as mentioned in our manuscript, the effects of IF on older mice were less profound than young adult mice. IF did not significantly prevent all abnormality examined. To clarify, we revised the text to claim the effects of IF on older mice were “mild” and “partial”.

4. On page 2, line 23, the statement “Dry AMD, also known as geographic atrophy, accounts for approximately 90% of all the AMD cases”, should be revised for accuracy. While dry AMD accounts for about 90% of AMD cases, geographic atrophy specifically refers to the later or advanced stage of dry AMD.

Response: We have revised the statement for accurate description of types of AMD, as follows: “Dry AMD accounts for approximately 90% of all AMD cases. Among these, about 10-20% of patients progress to geographic atrophy (GA), the advanced stage characterized by slowly progressive atrophy of the RPE accompanied by degeneration of the adjacent neural retina” (Page 3, Lines 2-5).

5. On page 8, line 2, the statement “the AL+NaIO₃ group exhibited significant pigment fragmentation and accumulation within the RPE layer”, should be supported with quantitative data when using the term “significant.”

Response: Thank you for your guidance in enhancing the clarity and precision of our manuscript. We have added a relative quantification analysis of pigment deposition in RPE flat mounts to support the term "significant" in the statement on Page 8, line 2. The quantitative data, now included in the supplementary materials (Supplementary Figure S3B-C, Page 4), confirms the statistically significant pigment accumulation and aggregation observed in the AL+NaIO₃ group compared to controls.

6. In Figure 5C, the label should be corrected to IF+NaIO₃.

Response: We apologize for the mistake. The label in Figure 5C has been corrected to "IF+NaIO₃". We have also thoroughly checked all figure labels and legends for similar issues.

7. On page 20, line 1, the statement “Immunostaining of retinal flat mounts revealed the morphological characteristics of ramified and activated microglia in the NaIO₃-challenged mice (Fig. 6B)”, please ensure whether the correct figure reference should be Figure 6C.

Response: Thank you for your careful review. Upon rechecking the figure order and corresponding text, we confirmed that the correct reference should indeed be Figure 6C instead of Figure 6B. In addition, we rearranged the order of the images during revision and annotated them accordingly in the text (Page 21, Line 5).

8. In Figure 6D, please ensure the y-axis label is correctly labeled as “IBA1+ intensity.”

Response: Thank you for identifying the mislabeling. This figure has been renumbered to Figure 6G during manuscript revision, and the y-axis label now correctly reads “IBA1+ intensity” (Page 22, Figure 6G).

Response to reviewers' comments:

Reviewer #1 (Remarks to the Author):

The authors have addressed most of the suggested revisions. However, I would like to raise a methodological concern regarding the quantification of fluorescence intensity. The use of integrated density as a readout is not ideal, as it is directly influenced by the size of the area measured—larger areas will inherently yield higher values, which may confound interpretation. I strongly recommend that the authors express the data as mean fluorescence intensity (MFI) after appropriate background subtraction in all panels involving fluorescence measures. This approach normalizes for area and will provide a more accurate representation of fluorescence signal, minimizing potential artifacts arising from variable region sizes.

Response: We agree with the reviewer that the integrated fluorescent density is not always an ideal or appropriate measurement for fluorescent signals. We have made revisions to use MFI instead for some experiments. Fluorescence in the revised manuscript are quantified in three ways as follows:

1) Some biological changes are manifested as changes in numbers of cells or fluorescent signals, including RPE cell numbers in Fig. 1A, TUNEL+ cells in Fig. 3H/7F, γ H2AX+ nuclei in Figure 4G. Therefore, the fluorescent signals were counted and presented as numbers of positive cells, which inherently avoids area-related biases.

2) Nile red intensity correlates with lipid droplet levels in the RPE cells. Similarly, fluorescent intensity of the lysosomal marker CD68 reveals phagocytotic activation of microglia. For these two types of signals, MFI is a more appropriate indication of the biological changes. We have replaced Fig. 2H and 6C using MFI with background subtraction.

3) Alterations in spatial distribution or area are intrinsic properties of certain biological changes. For example, Müller cells undergo distinct morphological changes upon activation, including somatic hypertrophy, process protrusion, end foot swelling, and so on (PMID: 21921569). While MFI of GFAP fluorescence captures some of the biological properties of Müller cells, integrated density is a more comprehensive readout revealing their morphological changes. In line with the field, integrated density was used to quantify the fluorescent signals of GFAP, PNA, Rhodopsin, and IBA1 [PMID: 33895485 (Rho/PNA), 35922863 (GFAP, IBA1), 37934489 (GFAP), 35347235 (IBA1)]. We share the reviewer's concerns about non-specific signals using integrated intensity as a readout. Therefore, we applied strict background subtraction and uniform thresholding across all samples to minimize non-specific signals, ensuring

that integrated density values reflect true biological changes rather than arbitrary area selection.

Reviewer #2 (Remarks to the Author):

Thank you to the authors for providing the revised manuscript. Please find additional comments below:

1. In Figure 1J and Figure S2B, could the authors clarify the detailed statistical methods used for comparing the number of head rotations at different frequency among four groups?

Response: We apologize for the confusion. OMR test is often presented as curves of head rotations versus cpd. And the difference between curves is statistically analyzed using two-way ANOVA to reveal the overall visual function (Fig. 1J and S2B). Visual acuity, cpd at which head rotation disappears, is also analyzed using two-way ANOVA to indicate the changes in visual limit (Fig. 1K and S2C). The statistical method has been included in the revised figure legends.

2. *Hprt1* shows a CT value around 28, suggesting relative low abundance in the samples. This makes it a less ideal choice for a house-keeping genes.

Response: As the reviewer noted, the relatively high Ct value (~28) indicates low abundance of *Hprt1* in our samples, which may compromise its reliability for normalization. In response to this concern, we have tested a few more reference genes for retinal samples, of which the ubiquitin gene *Ubc* was highly expressed with Ct values ~23. We have replaced *Hprt1* with *Ubc* as a reference gene. All target genes have been re-normalized using the geometric mean of the three reference genes (*Ubc*, *Gapdh*, and β -*actin*). The results have been updated in the main figures and tables of the manuscript. Additionally, the original Ct values for all genes are provided in Supplementary Excel 2.

3. In Figure S7A, the PCA reveals that the variance captured by PC1 and PC2 is primarily driven by heterogeneity within AL+NaIO₃ group. This within-group variability is also observed in Figure 5C. The authors should be cautious in interpreting global sample similarity. Additionally, the authors should check the sample-level QC (e.g., sample quality, sequencing quality, mapping quality) to rule out any technical artifacts that may contribute the presence of outlier sample.

Response: We understand the reviewer's concerns. We double-checked QC data for RNA sequencing, including metrics for sample quality, sequencing depth, and mapping efficiency, confirming no technical artifacts.

The observed variability in the AL+NaIO₃ group is probably attributed to variance between the biological replicates. While it is desirable to get uniform data between samples, there are intrinsic variations when disease modeling in animals, especially in acute models using chemical methods. Similar in-group variations are observed in NaIO₃-induced AMD rodent models (PMID: 36036255).

While uniform data provides power to identify more subtle changes, data with some variation like ours can still capture the top changes and provide the most significant changes in biological processes.

Reviewer #3 (Remarks to the Author):

Age-related macular degeneration (AMD) is the leading cause of blindness. No treatments are currently available for dry AMD, which affects 90% of AMD patients and only limited treatment options exist for wet AMD. Intermittent fasting (IF), a dietary intervention, has shown benefits for various diseases, including metabolic, cardiovascular, and neurodegenerative disorders. In this study, the authors investigated the effects of IF on AMD pathogenesis and its underlying mechanisms using a NaIO₃-induced AMD mouse model. The experimental data demonstrated that IF alleviated NaIO₃-induced damage by inhibiting glial cell activation and reducing ROS production. Furthermore, IF was also beneficial in older mice. The authors suggest that IF could serve as a potential therapeutic strategy for AMD and other retinal diseases associated with oxidative stress.

The study is unique, and the data clearly presented and supported the authors' hypothesis. The manuscript is well-written and easy to follow. The experiments were logically designed and meticulously conducted. I have only a few minor comments for the authors.

1. The NaIO₃ treatment appears to have an acute effect on the retina, which I assume is why the authors administered the drug at the end of the IF experiments. However, in the experimental paradigm shown in Figure 1A, it is unclear whether the mice remained on IF from the day of drug treatment to tissue collection (up to 21 days post-treatment) or if the observed effects could be attributed to the aftereffects of IF. Clarification on this point would strengthen the interpretation of the results.

Response: The diet regimes (AL or IF) were maintained after NaIO₃ injection until end of experiments. Figure 1A (Page 7) and Methods (Page 33, Lines 16-17) have been updated to clearly indicate the continuation of diet regimes. As the reviewer noted, NaIO₃ is considered to trigger acute retinal damage within 24 hours (PMID: 27551542). Therefore, IF-mediated protection of retinal damage is mainly attributed to preconditioning mechanisms (e.g., enhanced stress resilience prior to injury), although potential post-injury benefits cannot be ruled out.

2. In the Methods section, the authors stated that mice were reared under a 12:12 h light/dark cycle, with food added and removed at 5:30 PM. It would be helpful to specify the light on/off times to ensure clarity regarding food availability during the mice's active phase.

Response: Thank you for raising this important point. We employed an alternate-day fasting paradigm, in which food is available for 24 h then removed for 24 h. So, mice

have ad libitum access to food for one night (active phase) then no food for the following night. In the revised manuscript, we have added the following details to the Materials and Methods section (Page 32, Lines 9-12): “Mice were maintained under a 12:12 h light/dark cycle, with lights on from 7:00 AM to 7:00 PM. Food was provided daily at 5:30 PM (1.5 hours before the dark phase onset) and removed at the same time the following day to ensure ad libitum access during their active nocturnal period.”

3. In Figure 3B and 3C, the retinal diameter appears to be unusually large (10 mm) for mice. I recommend rechecking the data to ensure accuracy.

Response: We thank the reviewer for careful examination of our data. Upon re-examination, we realized an incorrect scale bar conversion (originally labeled 10 mm). We have recalibrated all images, and the corrected retinal diameter is approximately 5 mm. Figures 3B and 3C with revised annotations have been updated in the manuscript.

4. In Figure 4G, it is unclear how the authors interpret the expression of ROS in the RPE and NOX2 in the GCL. Providing a detailed explanation of their interpretation would enhance clarity.

Response: We appreciate the reviewer’s comment on the spatial inconsistency between NOX2 expression and ROS signal in the representative images. Similar concern was also raised during the last revision. We performed additional immunofluorescence of γ H2A.X (Page 16, Figure 4G), an indicator of oxidative DNA damage found in many pathological conditions including overproduction of ROS (PMID: 34073310). We also went through our original images of ROS, NOX2, γ H2A.X and cleaved caspase 3. As shown in figure below, signals of ROS, γ H2A.X, and cleaved caspase 3 in the AL+NaIO₃ retinal sections were all found primarily around the nuclear layers, especially inside or around the ONL region. NOX2⁺ signals were also found around the ONL region. Together, the four signals around the ONL layer indicate ROS overproduction and oxidative damage in NaIO₃-treated retina. We apologize that some of the images in our last version of manuscript were not representative. We have replaced them with more representative images. We thank the reviewer for prompting this re-evaluation for better presentation of our data.

5. In Figure 6, while the authors tried to present representative images alongside the quantification of the data, the layout is somewhat difficult to follow. I suggest reorganizing Figure 6 to improve the presentation.

Response: Thank you for your suggestion. We have replaced some panels and reorganized the layout following the reviewers' comments. In this version, the quantification charts are by the side of representative images for a better flow.

6. NaIO₃-induced damage was predominantly observed in the central retina/RPE in young mice, whereas in older mice, the damage appeared more peripheral. I wonder if the authors have any explanation or comments on this observation.

Response: We appreciate your insightful question regarding the spatial differences in NaIO₃-induced retinal damage between young and aged mice. In young mice, low-dose NaIO₃ primarily targets the central retina/RPE near the optic nerve, consistent with prior reports highlighting regional vulnerability in this area (PMID: 31932580, 30025075). In aged mice, this damage extends to peripheral regions, aligning with reports of expanded lesion areas in aged retinas under equivalent NaIO₃ exposure (PMID: 32826201). We speculate that this spatial expansion reflects age-associated declines in antioxidant defenses and/or amplified oxidative stress sensitivity in aging retinal tissues.

7. In the Discussion (page 27, lines 14-18), the authors suggest that the older mice exhibited greater retinal/RPE damage from NaIO₃ treatment due to increased susceptibility to oxidative stress. However, I believe no data has been provided to support this claim. It would be helpful to include supporting evidence or clarify this point.

Response: We thank the reviewer for highlighting the need for caution in interpreting age-related differences. We have revised the discussion (Page 27, Lines 14-21) to remove unsupported claims about oxidative stress susceptibility. The updated text now states: “Our data reveal age-related attenuation of IF-mediated protection against NaIO₃-induced retinal damage. This may reflect either diminished IF efficacy in aged mice or their heightened vulnerability to NaIO₃ toxicity, as evidenced by exacerbated middle and periphery RPE degeneration (Fig.S4, Fig.7A-C). While oxidative stress pathways likely contribute to this disparity, mechanistic distinctions between age-specific IF responsiveness and NaIO₃ sensitivity remain unresolved. Targeted investigations comparing molecular responses to IF and toxin challenge across age cohorts will clarify these interacting factors.” This revision emphasizes the observational nature of our findings while acknowledging the need for future validation.

8. If the data is available, it would be nice to include physiological and morphological parameters such as body weight, food consumption, OMR, and retinal thickness in older mice.

Response: We appreciate the reviewer’s constructive suggestion to expand the physiological and morphological characterization of aged mice. In accordance with this recommendation, we have incorporated retinal thickness data from 16-month-old mice into the revised manuscript (Supplementary Figure S9). Regretfully, parameters such as body weight, food consumption, and OMR were not systematically recorded in the original experimental design. We acknowledge the importance of these parameters for comprehensive analysis and will pursue in future investigations to further validate our findings.

Response to reviewers' comments:

Reviewer #1 (Remarks to the Author):

As noted in my previous comment, the use of integrated density to quantify marker expression in retinal areas is not a reliable approach, even when background subtraction is applied. This method combines signal intensity with the size of the measured area, which is particularly problematic in the retina, where fluorescence intensity can vary substantially depending on the extent and anatomical region selected.

This represents a significant methodological limitation. I strongly recommend that the authors reanalyze their retinal data using mean fluorescence intensity (MFI), which provides a more accurate and standardized measure of expression per unit area. Doing so would greatly enhance the robustness, reproducibility, and interpretability of the results.

Response: We sincerely appreciate your critical feedback regarding the quantification of marker expression in retinal areas. In response to your recommendation, we have replaced the integrated fluorescence intensity with mean fluorescence intensity (MFI) for quantification of the retinal images. This approach ensures that expression levels are quantified per unit area, eliminating the influence by regional size variations. The detailed methodology for MFI analysis has been included in the "Methods" section of the revised manuscript (please refer to Pages 36-37, Section 4.8 and 4.9).

Importantly, while the transition to MFI resulted in changes in the magnitude of differences and/or statistical significance for a minority of the datasets, the overall conclusions of the study remain unchanged. The updated data have been incorporated throughout the manuscript, including related figures and tables. These revisions now provide more robust and reproducible results, as you suggested, without altering the study's fundamental interpretations. We believe this approach significantly enhances the methodological rigor of our findings.